# Forecasting China's shipping indices based on modal decomposition and optimized deep learning integrated model

**Yuye Zou** , **Yingyu Liu, Guangnian Xiao***

College of Economics and Management, Shanghai Maritime University, Shanghai, China

* gnxiao@shmtu.edu.cn

## Abstract

This study proposes an innovative hybrid forecasting model, VMD-CPSO-BiLSTM, which significantly enhances the prediction accuracy of shipping indices in China's maritime sector. The model employs a sophisticated three-phase methodology: (1) decomposition through Variational Mode Decomposition (VMD) to extract multiple intrinsic mode functions (IMFs) from the original time series, effectively capturing its nonlinear and complex patterns; (2) optimization using a Chaotic Particle Swarm Optimization (CPSO) algorithm to fine-tune the Bi-directional Long Short-Term Memory (BiLSTM) network parameters, thereby improving both predictive accuracy and model stability; and (3) integration of predictions from both high-frequency and low-frequency components to generate comprehensive final forecasts. Through extensive empirical validation using key Chinese shipping indices, our proposed model demonstrates superior performance compared to conventional single deep learning models and other hybrid approaches. The results indicate that VMD-CPSO-BiLSTM effectively addresses critical challenges in time series forecasting, including nonlinearity, non-stationarity, and multi-scale characteristics. The developed model offers substantial practical value as a reliable forecasting tool for shipping market trends, providing industry stakeholders with enhanced decision-making support for strategic planning and operational management. Its robust performance and methodological innovation contribute significantly to the field of maritime economics and financial time series analysis.

## Introduction

As one of the largest merchandise traders in the world, China's shipping industry has a significant impact on global trade flows. The Chinese shipping indices reflect the activity level of the domestic shipping market, the changes in demand for shipping capacity, and the trends in freight rate fluctuations. With the advancement of China's "Belt and Road" initiative and the continuous growth of domestic and international trade and investment, China's shipping industry has gradually taken on a

**Data availability statement:** There are no legal or ethical restrictions on sharing the de-identified data from this study. The minimal anonymized dataset required to replicate all study findings has been uploaded to the Figshare repository. DOI: 10.6084/m9.figshare.30616022.

**Funding:** This work was supported by the National Natural Science Foundation of China (No. 12101393) awarded to Z.Y.Y.

**Competing interests:** The authors have declared that no competing interests exist.

more important strategic position in the global market, especially in areas such as global container transport and energy transportation, where it plays a key role. Therefore, analyzing China's shipping index not only helps in understanding the changes in the domestic shipping market but also provides a reference for the trends in the international shipping market.

Predicting China's shipping indices holds significant practical value. First, accurate forecasting can help shipping companies make more informed operational decisions, optimize route layouts, schedule resources, and adjust shipping capacity to respond to market demand fluctuations. Second, for international trade and logistics companies, forecasting the shipping index helps assess freight rate trends and predict changes in transportation costs, thereby optimizing supply chain management. Additionally, the government and relevant policymakers can adjust policies in a timely manner based on changes in the shipping index, promoting the healthy development of the shipping industry and enhancing international competitiveness. By scientifically forecasting the shipping index, market risks can be effectively mitigated, and China's shipping industry can strengthen its influence and competitive advantage in the global market.

The prediction of shipping indices can be based on different econometric models, depending on the characteristics and complexity of the data. Commonly used models include ARIMA for stationary data [1,2], VAR for considering multiple influencing factors [3,4], GARCH for data with higher volatility [5–7], ECM for considering long-term equilibrium relationships and short-term dynamic adjustments [8,9], and panel data models for multidimensional data analysis [10,11]. With the arrival of the big data era, the exponential growth of data and significant advancements in computational power have laid the foundation for the rapid rise of the artificial intelligence industry. This progress has provided innovative solutions to the challenges of financial time series forecasting. As a result, machine learning techniques have gradually replaced traditional econometric models, becoming the primary tools for predicting financial trends and behaviors.

The shipping index typically exhibits complex temporal characteristics and is influenced by various external factors, such as economic fluctuations and changes in market demand. These factors make the variations in the shipping index not only time-dependent but also contain periodic fluctuations of different frequencies. Traditional time series analysis methods and single neural network models often struggle to fully extract key features from such complex sequences. Time-frequency analysis methods, such as Fourier transform and wavelet transform, can decompose time series signals into components of different frequencies, thereby revealing hidden periodic fluctuations and short-term trends in the data. By applying time-frequency analysis to the shipping index, it is possible to better capture dynamic features at different time scales, improving the accuracy and reliability of predictions. [12] and [13] predicted the shipping index using wavelet transform and support vector machines (SVM), respectively. [14] combined wavelet transform and neural networks to study freight rate fluctuations. In time-frequency analysis methods, wavelet transform improves upon traditional Fourier transform by offering time-frequency localization

characteristics, which makes it effective in analyzing non-stationary signals and capturing local variations in the signal across different time scales. However, wavelet transform has an issue with uneven resolution, especially with a significant resolution difference between low and high frequencies, which may lead to information loss or inaccuracies. As an improvement, empirical mode decomposition (EMD) has emerged. It can adaptively decompose signals into multiple IMFs, overcoming the resolution limitations of wavelet transform and making it suitable for handling nonlinear and non-stationary complex signals, thereby enhancing the accuracy and flexibility of time-frequency analysis [15–17]. Variational mode decomposition (VMD), a novel signal decomposition method proposed by [18], decomposes complex signals into modes with different central frequencies and limited bandwidths through variational optimization. Compared to traditional EMD and its improved methods (such as EEMD, CEEMDAN), VMD introduces bandwidth constraints in the frequency domain, avoiding issues like mode mixing and endpoint effects, and offering better noise robustness and stability. [19] applied VMD to decompose the SCFI index and combined it with an extreme learning machine (ELM) model for prediction, achieving remarkable results.

In recent years, machine learning and deep learning models have been widely applied to forecast shipping indices, primarily including SVM [20,21] and fuzzy neural networks [22]. Traditional machine learning models rely on manually extracted features, which are often constrained by expert knowledge and the complexity of high-dimensional data, leading to limited performance when handling large-scale data. In contrast, deep learning automatically learns feature representations, enabling it to handle more complex tasks and large datasets with stronger generalization ability and accuracy. To improve prediction accuracy, scholars have used artificial neural networks (ANN) [23,24] and long short-term memory networks (LSTM) [25,26]. Although the LSTM model has demonstrated strong capabilities in time series forecasting, the standard LSTM only predicts based on past information. In comparison, BiLSTM processes information from both forward and backward directions (from future to past), allowing the model to capture more comprehensive contextual relationships within the sequence, thereby enhancing prediction performance and accuracy.

The training process of machine learning models typically involves the selection of multiple hyperparameters, such as learning rate, regularization coefficient, number of layers, and the number of neurons. Traditional hyperparameter tuning methods, such as grid search and random search, although widely used, often involve a large amount of computation and are prone to getting stuck in local optima. Particle swarm optimization (PSO), as a global optimization algorithm, simulates the process of particles sharing information and cooperating to find the optimal solution in the solution space, effectively optimizing the hyperparameters and weights of the model, thereby enhancing model performance. Machine learning models optimized by PSO can find more suitable hyperparameter configurations, improving prediction accuracy, accelerating the training process, and enhancing the model's generalization ability, especially in handling complex problems, significantly improving model performance. [27] used the PSO-LSTM model to predict the global six major stock indices, and the results showed that the optimized PSO-LSTM model outperformed the traditional LSTM model in prediction effect. [28] used the Twofold PSO-LSTM to predict the Baltic Dry Index, Dirty Tanker Index, and Container Index during the 2010-2022 crisis period, and the prediction effect was better than that of other models. However, PSO is prone to getting stuck in local optima during the search process, especially in high-dimensional and complex problems, where the particle swarm may converge prematurely. By introducing chaotic systems, the nonlinearity and randomness of particle velocity updates are enhanced, further improving the exploration ability of the particle swarm in the solution space and avoiding the problem of local optima. The CPSO can maintain the diversity of the particle swarm and enhance the global search ability, thereby improving the convergence speed and optimization accuracy of PSO [29].

In general, existing literature on shipping index forecasting models mainly focuses on traditional econometric models or simple machine learning models. Research on combining VMD with deep learning models for shipping market prediction is relatively scarce. [30] applied the hybrid CEEMD-PSO-BiLSTM model to predict the shipping index. However, the predictive performance of the CEEMD-PSO-BiLSTM model is inferior to that of the proposed VMD-CPSO-BiLSTM

model. First, VMD demonstrates advantages over CEEMD in terms of theoretical rigor, modal purity, and resistance to end effects. More importantly, VMD's critical parameter K can be interactively optimized with our algorithm, whereas CEEMD's parameters are difficult to effectively optimize. Second, CPSO addresses the issue of premature convergence commonly found in standard PSO. With enhanced global search capability and superior convergence precision, it enables the identification of an optimal set of hyperparameters for the BiLSTM network. Finally, VMD-CPSO-BiLSTM constitutes a highly synergistic system. CPSO not only optimizes the hyperparameters of BiLSTM but also simultaneously adjusts the modal number K of VMD. In contrast, within the CEEMD-PSO-BiLSTM framework, PSO is typically only employed for BiLSTM parameter optimization. The decomposition parameters of CEEMD are preset manually and fixed, incapable of collaborative optimization with subsequent modules. Based on the above advantages, the VMD-CPSO-BiLSTM model is expected to comprehensively outperform the comparative model in predictive accuracy, stability, and generalization capability.

Specially, the VMD-CPSO-BiLSTM model first decomposes the shipping index using the VMD, then reconstructs and reassembles the IMFs. Next, the CPSO-BiLSTM model is constructed to predict each subsequence, and finally, the predicted results of each subsequence are integrated to obtain the final forecast. In this study, four major shipping indices of the Chinese shipping market—namely the China Containerized Freight Index (CCFI), the Shanghai Containerized Freight Index (SCFI), the China Dry Bulk Freight Index (CDBI), and the China Coastal Bulk Freight Index (CBFI)—are selected as test cases. The proposed model is compared with existing standalone deep learning models and various hybrid models based on these methods to validate the effectiveness of the proposed approach. The innovations and contributions of this paper are primarily reflected in the following two points:

1. Proposing an innovative hybrid VMD-CPSO-BiLSTM model.

This paper proposes an innovative shipping index prediction model by combining VMD, CPSO and BiLSTM. VMD effectively decomposes the non-stationary shipping index signal into multiple IMFs with independent frequency components, avoiding the mode mixing problem in traditional EMD methods. CPSO, by introducing chaotic mechanisms, enhances the global search ability of the Particle Swarm Optimization algorithm in high-dimensional optimization spaces, thereby better optimizing the hyperparameters of BiLSTM. BiLSTM can capture both long-term and short-term dependencies in time-series data, further improving the accuracy of predictions. The integration of these models effectively overcomes the limitations of traditional methods in handling complex time-series data and provides a more accurate prediction tool.

2. Improving shipping index prediction accuracy and broad industry applications.

The application value of this paper lies in significantly improving the accuracy of shipping index prediction while providing strong methodological support for predicting nonlinear and non-stationary time-series data in other fields. Predicting the shipping index has significant economic implications, helping relevant businesses and policymakers make more accurate market judgments. Through the VMD-CPSO-BiLSTM model, each frequency component of the shipping index is precisely modeled, solving the complexity issues that traditional prediction methods cannot handle. In addition to shipping indices, this method has strong scalability and can be applied to other fields such as stock market prediction, energy demand forecasting, and more, demonstrating its broad practical application potential, especially in handling time-series data with complex patterns and fluctuating trends.

The rest of this paper is organized as follows: Section Materials and methods gives the principles of the sub-models used in this article and the construction process of the VMD-CPSO-BiLSTM combined model. Section Data provides a descriptive analysis of the original data of four China's shipping indices. In Section Results, we present the empirical analysis results, comparing the prediction outcomes of single models and combined models. Section Conclusion and discussion summarizes the main findings of this paper and provides directions for future research. Section Suggestion gives some suggestions.

## Materials and methods

### VMD method

VMD is an advanced signal processing method widely used in the analysis and decomposition of complex signals. Its core idea is to decompose the original signal into several IMFs, where each IMF represents a component of the signal with distinct frequency and amplitude characteristics. Traditional time-series decomposition methods (such as EMD) are often affected by the mode mixing problem, whereas VMD effectively avoids this issue by introducing the concept of frequency shift. The basic steps of VMD involve first constructing a constrained variational problem, then transforming it into an unconstrained variational problem, and finally finding the optimal solution through an iterative method. The specific decomposition steps are as follows:

**Step 1**: Construction of variational problems.

The variational problem is constructed as a constrained model to minimize the total bandwidth of all modes. For each mode, the analytic signal is generated via Hilbert transform, frequency-shifted to baseband, and its bandwidth is estimated through the squared $L_2$-norm of the demodulated signal's gradient. The constrained formulation is expressed as (1) and (2).

$$\min_{u_k, w_k} \left\{ \sum_{k=1}^{K} \left\| \frac{\partial}{\partial t} \left[ \left( \delta(t) + \frac{\pi t}{j} * u_k(t) \right) \right] e^{-jw_k t} \right\|_2^2 \right\} \tag{1}$$

$$s.t. \sum_{k=1}^{K} u_k(t) = x(t), \tag{2}$$

where K represents the total number of components. $\delta(t)$ is the Dirac function. $j$ is an imaginary unit. * is volumetric cumulant computing. $w_k$ is the center frequency of the K-th component. $u_k$ is the function of each mode.

**Step 2**: Transformation into an unconstrained problem.

To solve the above constrained problem, it is transformed into an unconstrained one by introducing both a quadratic penalty term and Lagrangian multipliers $\lambda$. The augmented Lagrangian function $\mathcal{L}$ is defined as the following Eq (3).

$$\mathcal{L}(u_k, w_k, \lambda) = \alpha \sum_{k=1}^{K} \left\| \frac{\partial}{\partial t} \left[ \left( \delta(t) + \frac{\pi t}{j} * u_k(t) \right) \right] e^{-jw_k t} \right\|_2^2$$

$$+ \left\| f(t) - \sum_{k=1}^{K} u_k(t) \right\|_2^2 + \left\langle \lambda(t), f(t) - \sum_{k=1}^{K} u_k(t) \right\rangle, \tag{3}$$

where $\alpha$ is the penalty parameter that balances the reconstruction fidelity and the bandwidth constraint.

**Step 3**: Solution via the alternate direction method of multipliers.

The optimal solution of the unconstrained variational problem is sought using the Alternate Direction Method of Multipliers (ADMM). This involves iteratively updating each variable $u_k$, $\omega_k$ and $\lambda$ in a alternating manner while holding others fixed, until convergence is achieved. The iterative process is shown as Eqs (4)–(6).

$$\hat{u}_k^{n+1}(w) = \frac{\hat{f}(w) - \frac{\sum_{i \neq k} \hat{u}_i^n(w) + \hat{\lambda}^n(w)}{2}}{1 + 2\alpha(w - w_k)^2}, \tag{4}$$

$$w_k^{n+1} = \frac{\int_0^\infty w |\hat{u}_k^{n+1}(w)|^2 dw}{\int_0^\infty |\hat{u}_k^{n+1}(w)|^2 dw}, \tag{5}$$

$$\lambda^{n+1}(w) = \lambda^n(w) + \gamma\left(\hat{f}(w) - \sum_{i=1}^{K} \hat{u}_i^{n+1}(w)\right), \tag{6}$$

where $\gamma$ is the update parameter for the dual ascent. Repeat the above steps until the iterative constraint condition $\sum_{k=1}^{K} \|\hat{u}_k^{n+1} - \hat{u}_k^n\|^2 / \|\hat{u}_k^n\|^2 < \varepsilon$ is met.

## CPSO algorithm

PSO is an optimization algorithm based on swarm intelligence in the solution space, while the velocity determines the direction and speed at which the particle moves through the space [31]. Particles adjust their flight direction and speed based on both their individual experiences and the experiences of the swarm, gradually converging toward the optimal solution. Specifically, each particle records two key pieces of information during its flight: the individual best solution ($P_i$) and the global best solution ($P_g$). In each iteration, particle i updates its position $x_i = (x_{i1}, x_{i2}, \cdots, x_{id})$ and velocity $v_i = (v_{i1}, v_{i2}, \cdots, v_{id})$ based on its own experience and the swarm's experience. The formulas for updating the particle's velocity and position are defined as the following Eqs (7) and (8).

$$v_i^{k+1} = wv_i^k + c_1r_1(P_i - x_i^k) + c_2r_2(P_g - x_i^k), \tag{7}$$
$$x_i^{k+1} = x_i^k + v_i^{k+1}, \tag{8}$$

where $v_i^k$ is the velocity of particle i at iteration k. $x_i$ is the position of particle i at iteration k. w is the inertia weight, used to control the inertia of the particle's movement. $c_1$ and $c_2$ are the learning factors. $r_1$ and $r_2$ are random numbers uniformly distributed between [0,1].

PSO is prone to issues such as getting stuck in local optima and having slow convergence during the optimization process. CPSO is an improved version of the PSO algorithm, with its core idea being the introduction of a constriction factor, which controls the velocity update formula of the particles, thereby improving the algorithm's convergence speed and global search ability. The velocity update formula is as the following Eq (9).

$$v_i^{k+1} = xv_i^k + c_1r_1(P_i - x_i^k) + c_2r_2(P_g - x_i^k), \tag{9}$$

where x is the constriction factor, which is used to limit the magnitude of the particle's velocity, preventing the particle from moving too quickly through the search space and thereby potentially missing the global optimum. The formula for calculating the constriction factor x is as the following Eq (10).

$$x = \frac{2 \cdot |2 - \phi - \sqrt{\phi^2 - 4\phi}|}{\phi^2 - 4\phi}, \tag{10}$$

where $\phi = c_1 + c_2$ and $\phi > 4$. By selecting an appropriate value for $\phi$, the constriction factor can be effectively adjusted, thereby balancing the algorithm's global search capability and local search ability.

## BiLSTM model

BiLSTM is a special type of LSTM structure. In traditional LSTM, information is passed in one direction, whereas BiLSTM simultaneously considers both forward and backward information. Specifically, BiLSTM consists of two LSTM layers: a forward LSTM and a backward LSTM. The outputs of these two LSTMs are concatenated in the hidden layer, forming a

feature representation that contains both forward and backward information. This design allows BiLSTM to better capture long-term dependencies in the sequence and utilize both the past and future context. The process of BiLSTM is as follows:

**Step 1**. BiLSTM first feeds the input data (such as a text or time series) into two LSTM networks, one for the forward direction and one for the backward direction.

**Step 2**. In the forward LSTM, information is gradually passed from the beginning of the sequence to the end; in the backward LSTM, information is passed from the end of the sequence to the beginning.

**Step 3**. After processing by both LSTMs, BiLSTM concatenates or performs a weighted average of the outputs from both directions to obtain the final sequence representation.

As shown in Fig 1, this model can be viewed as a deep neural network with two layers. The input to the first layer is located on the left side of the network structure, representing the sentence-initial input in natural language processing; the input to the second layer is located on the right side, representing the sentence-final input. After the calculations performed by these two hidden layers, two output results are obtained. The mathematical expression is as the following Eqs (11)–(13).

$$h_t = f(\omega_1 x_t + \omega_2 h_{t-1}), \tag{11}$$
$$h'_t = f(\omega_3 x_t + \omega_5 h'_{t+1}), \tag{12}$$
$$o_t = g(\omega_4 x_t + \omega_6 h'_t). \tag{13}$$

## Construction of combined VMD-CPSO-BiLSTM model

The shipping index exhibits characteristics of non-stationarity, non-linearity, and high complexity, making it difficult for a single deep learning forecasting method to accurately capture its main features. First, the VMD is used to decompose the time series of the shipping index into IMFs, where each IMF represents different frequency components of the signal. Next, the CPSO is applied to optimize the hyperparameters of the BiLSTM, enhancing the model's global search ability and preventing it from getting trapped in local optima. Then, the BiLSTM is used to model and forecast each IMF, as it

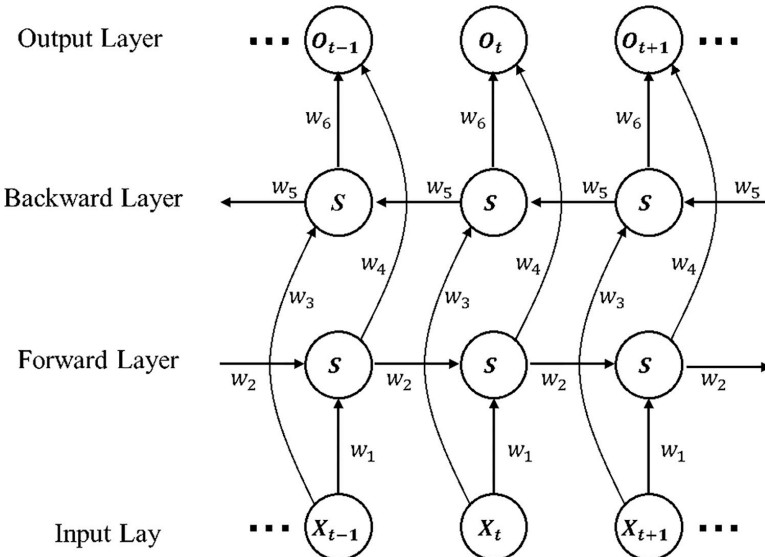

**Fig 1**. BiLSTM model structure diagram [30].

can capture both long- and short-term dependencies in the time series. Finally, the predictions from each IMF are combined using a weighted or fusion method to integrate the predicted values of all components, resulting in the final forecast of the shipping index. By combining decomposition, optimization, and deep learning, this approach can effectively improve the forecasting accuracy of the shipping index. The VMD-CPSO-BiLSTM network structure is given in Fig 2. The specific modeling steps are as follows:

**Step 1**. Data Preprocessing.

First, missing values and outliers in the data are removed, and missing values are filled using interpolation or other methods to ensure data integrity. Next, standardization or normalization techniques are applied to the data, transforming it into a uniform scale to prevent the influence of varying data ranges on model training.

**Step 2**. Variational Mode Decomposition (VMD).

The VMD algorithm decomposes the shipping index time series signal into multiple frequency components, with each IMF representing a signal from a different frequency band. The advantage of VMD is that it avoids the modal mixing problem seen in traditional Empirical Mode Decomposition (EMD), and can more accurately capture the multi-scale characteristics of the signal. The resulting IMFs serve as the input for subsequent models, aiding in improving forecasting accuracy.

**Step 3**. Chaotic Particle Swarm Optimization (CPSO).

CPSO is applied to optimize the hyperparameters of the BiLSTM model, ensuring better training and prediction of the IMF subsequences after VMD decomposition. Specifically, the objective of CPSO optimization is to adjust key parameters of the BiLSTM, such as learning rate, number of hidden nodes, number of layers, etc. Through multiple iterations of particle position and velocity updates, CPSO finds the optimal hyperparameter combination, thereby enhancing the performance of the BiLSTM model.

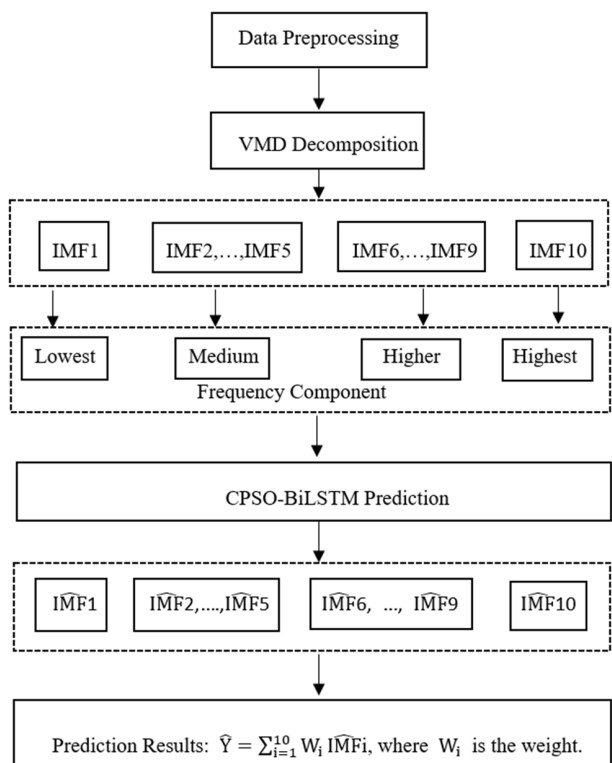

**Fig 2**. **VMD-CPSO-BiLSTM network structure.**

**Step 4**. Bidirectional Long Short-Term Memory Network (BiLSTM).

BiLSTM is used to model and predict each IMF subsequence after VMD decomposition. By utilizing both forward and backward propagation, BiLSTM can capture both long- and short-term dependencies in the time series data, making it suitable for handling the complex volatility in shipping index data. The trained BiLSTM model generates predictions for each IMF, reflecting the potential future trends of the shipping index.

**Step 5**. Results integration and forecast output.

The predictions from each IMF generated by the BiLSTM are integrated to form the final forecast of the shipping index. Typically, a weighted summation method is used to combine the predictions of each IMF, resulting in the overall forecast. Additionally, any residuals remaining from the VMD decomposition are predicted separately and combined with the IMF predictions. Through this multi-layer integration, the final forecast more accurately reflects the future trend of the shipping index, providing robust support for practical applications.

## Data

This article focuses on China's four major shipping indices: CCFI, SCFI, CDBI, and CBFI. The CCFI reflects the overall freight rate level in the Chinese export container shipping market, and is particularly significant for guiding import and export trade. The SCFI focuses on container freight rates at the Port of Shanghai and is an indicator of market trends at Shanghai and its surrounding ports. The CDBI reflects freight rate trends in China's dry bulk shipping market, especially offering valuable reference for the transportation of bulk commodities such as coal and ores. The CBFI specializes in the coastal bulk freight market, revealing changes in freight rates within China's coastal shipping market. Researching these shipping indices helps provide a deeper understanding of supply and demand fluctuations and freight rate volatility in the shipping market, offering valuable decision-making support to both the government and businesses. The data comes from the Ministry of Transport of the People's Republic of China (https://www.gov.cn/bumenfuwu/moc/), the Shanghai Shipping Exchange ( https://www.sse.net.cn/), China Economic Information Network (https://www.cei.cn/), and Clarkson Shipping Intelligence Network (Clarkson SIN) (https://sin.clarksons.net/).

From Fig 3, it can be seen that: (1) Before 2020, the CCFI experienced minor fluctuations. However, in 2020, the COVID-19 pandemic disrupted the global supply chain, reducing transportation demand and causing a sharp drop in the CCFI. By 2021, as China's economy recovered and global trade rebounded, the index surged. In 2022-2023, the index fluctuated due to ongoing pandemic effects, unstable supply chains, and geopolitical conflicts.

(2) The CDBI saw a decline from 2014 to 2016 due to the global slowdown, China's economic restructuring, and shifts in trade. From 2017 to 2019, it rose gradually, boosted by infrastructure investments and global recovery. The pandemic caused a sharp drop in 2020, but the index surged in 2021 with China's recovery. In 2022, it reached a historic high due to supply chain tightness and rising transportation costs. From 2023 to 2024, it fluctuated due to slower global growth and geopolitical tensions.

(3) The SCFI was stable from 2010 to 2015 but declined in 2016 due to global economic changes. It saw mild fluctuations from 2017 to 2019, indicating slower trade growth. The pandemic caused a sharp drop in 2020, followed by a surge in 2021 with the global recovery. In 2022, the SCFI peaked at over 8,000. From 2023 to 2024, it fluctuated due to slower growth and geopolitical tensions.

(4) From 2006 to 2008, the CBFI rose significantly, driven by global growth and strong demand for bulk cargo. After the 2008 financial crisis, it dropped sharply, and by 2016, it reached a low point. From 2016 to 2020, the index slowly recovered, influenced by China's economy and the Belt and Road Initiative. The pandemic caused a drop in 2020, but the index rebounded in 2021. After 2022, it declined due to reduced demand, economic uncertainty, and the Russia-Ukraine conflict.

From Table 1, it can be seen that the mean values of the four shipping indices are large, reflecting the overall high level of China's shipping market. However, their SD (standard deviations) are also large, indicating significant volatility

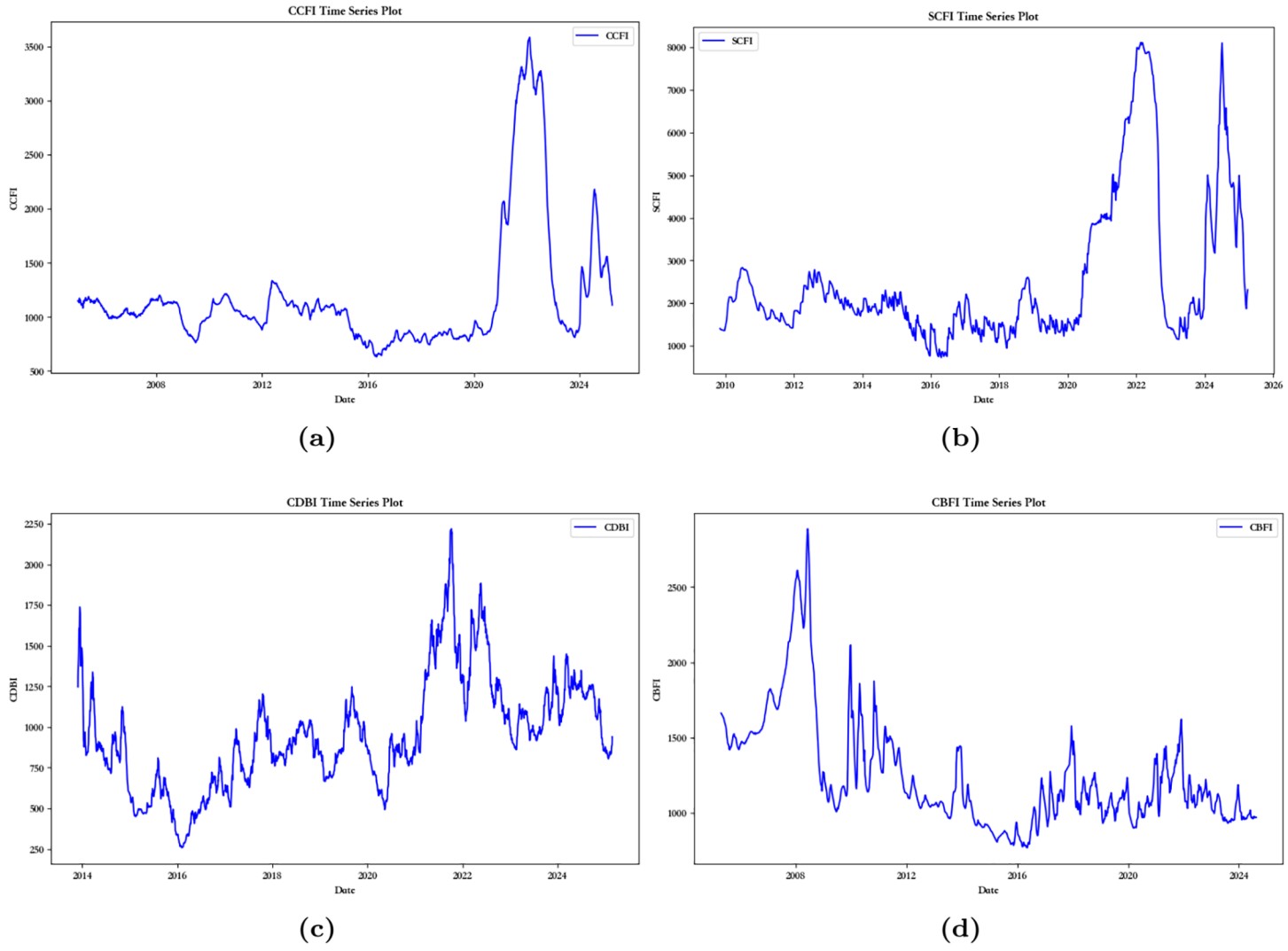

Fig 3. **The time series chart of China's shipping indices:** (**a**) The time series chart of CCFI. (**b**) The time series chart of SCFI. (**c**) The time series chart of CDBI. (**d**) The time series chart of CBFI.

Table 1. **Descriptive statistics of China's shipping indices.**

| Indices | Mean | SD | Kurtosis | Skew | Min | Max | ADF | $p$ |
|---|---|---|---|---|---|---|---|---|
| CCFI | 1183.80 | 582.60 | 6.29 | 2.60 | 632.36 | 3587.91 | −3.87 | 0.12 |
| SCFI | 2542.26 | 1705.07 | 2.54 | 1.83 | 725.00 | 8117.00 | −2.71 | 0.07 |
| CDBI | 950.78 | 338.39 | 0.39 | 0.58 | 259.06 | 2219.79 | −2.44 | 0.13 |
| CBFI | 1254.34 | 368.78 | 3.39 | 1.73 | 771.01 | 2886.92 | −2.41 | 0.14 |

in the indices. The positive skewness and kurtosis coefficients suggest that the four shipping indices do not follow a normal distribution. The CCFI has the highest kurtosis and skewness coefficients, while the CDBI has the lowest kurtosis and skewness coefficients. This suggests that the original data of the CCFI shipping index exhibits strong skewness and a sharp distribution, potentially containing more extreme fluctuations or outliers, leading to higher volatility. In contrast, the original data of the CDBI shipping index shows smaller skewness and kurtosis, with relatively stable fluctuations in the

data. The Augmented Dickey-Fuller (ADF) test statistic and its corresponding p-value are employed to determine whether a time series is stationary. A smaller p-value indicates stronger evidence against the null hypothesis of a unit root, suggesting that the series is more likely to be stationary. At the significance level of 0.01, the p-values for all four shipping indices exceed this threshold, leading to the failure to reject the null hypothesis. This provides statistical evidence that these four shipping index series are non-stationary. By using modal decomposition, the shipping indices can be broken down into IMFs, effectively separating long-term trends from short-term fluctuations, reducing noise interference, and improving the stability and accuracy of predictions.

## Results

### Evaluation criteria

In the model evaluation process, a set of performance metrics is typically used to ensure a more comprehensive and objective assessment of the model's predictive capabilities. These metrics reflect the differences between the model's predicted values and the actual observed values from various perspectives, thereby helping researchers evaluate the model's accuracy and reliability. Four commonly used performance evaluation metrics during model testing include mean squared Eerror (MSE), root mean square error (RMSE), mean absolute percentage error (MAPE) and mean absolute error (MAE). The specific formulas are as follows. Let $y_t$ represent the actual values, $\hat{y}_t$ represent the predicted values, and $n$ be the number of samples.

1. Mean square error (MSE). MSE is a commonly used metric for assessing the magnitude of the difference between predicted values and actual observations. It quantifies the accuracy of the model's predictions by calculating the average of the squared prediction errors. The specific calculation method is as follows:

$$\text{MSE} = \frac{1}{n}\sum_{t=1}^{n}(y_t - \hat{y}_t)^2.$$

2. Root mean square error (RMSE). RMSE is a metric used to measure the deviation between the model's predicted values and the actual values. The calculation formula is as follows:

$$\text{RMSE} = \sqrt{\frac{1}{n}\sum_{t=1}^{n}(y_t - \hat{y}_t)^2}.$$

3. Mean Absolute Percentage Error (MAPE). MAPE is a metric used to measure the accuracy of a predictive model by calculating the average absolute percentage difference between the predicted values and the actual values. The formula for MAPE is:

$$\text{MAPE} = \frac{1}{n}\sum_{t=1}^{n}\frac{y_t - \hat{y}_t}{y_t} \times 100\%.$$

4. Mean Absolute Error (MAE). MAE measures the average of the absolute differences between the model's predicted values and the actual values. The calculation formula is as follows:

$$\text{MAE} = \frac{1}{n}\sum_{t=1}^{n}|y_t - \hat{y}_t|.$$

5. Coefficient of determination $R^2$. $R^2$ is a statistical measure that explains how well the independent predictors in a regression model can predict the dependent variable. It is calculated as:

$$R^2 = 1 - \frac{SS_{error}}{SS_{total}},$$

where $SS_{error}$ is the sum of squared residuals. $SS_{total}$ is the total sum of squares.

## Data preprocessing

1. Missing data processing.

Missing data can affect model accuracy, so it is important to choose appropriate methods for handling it. Common approaches include deleting samples or features with missing values or filling the missing values, such as using the mean or methods like random forest regression. In this study, the missing data rate is 1.3%, and forward filling is used to handle the missing data, which involves filling a missing value with the most recent non-missing value before it. This method is particularly suitable for time series data, as time series typically exhibit temporal continuity.

2. Data standardization.

Data normalization involves mapping the values of variables to a specified range to reduce dimensionality. Standardizing the data can eliminate the impact of differing units across variables, ensuring that each variable has an equal influence on the objective function. It also enhances the model's learning efficiency and predictive accuracy. Therefore, data normalization is essential. There are various methods for data normalization, such as normalization and the Z-score method. In this paper, we adopt the Min-Max normalization method, and its formula is as follows.

$$z_i = \frac{x_i - x_{min}}{x_{max} - x_{min}},$$

where $z_i$ represents the result after Min-Max normalization, $x_i$ is the original sample data, $x_{min}$ is the minimum value in the original sample data, and $x_{max}$ is the maximum value in the original sample data. After Min-Max normalization, the data is scaled to fall within the range of [0, 1].

3. Dataset splitting.

Dataset splitting is a crucial step in machine learning and data analysis, aimed at dividing the dataset into different subsets for model training and testing. By properly splitting the dataset, we ensure that the model receives enough information during training to learn effectively, and we can evaluate its performance on new data using validation and test sets. This, in turn, enhances the model's reliability and generalization ability. In this paper, the first 80% of the data is used as the training set, while the remaining 20% is used as the test set. The division of the data set is shown in Table 2.

## VMD processing

VMD aims to decompose a complex signal into several Intrinsic Mode Functions (IMFs) with limited bandwidth, each centered around a specific frequency. The key idea is to minimize the sum of modal bandwidths while ensuring their sum

**Table 2**. The training set and test set of the shipping indices.

| Indices | Training set ($n_1$) | Testing set ($n_2$) | Unit | $n$ |
|---|---|---|---|---|
| CCFI | 2005.01-2022.03 (716) | 2022.04-2025.03 (307) | week | 1023 |
| SCFI | 2009.11-2023.03 (617) | 2023.04-2025.03 (154) | week | 771 |
| CDBI | 2013.11-2025.02 (2180) | 2023.08-2025.02 (545) | day | 2725 |
| CBFI | 2015.04-2022.01 (730) | 2022.02-2024.08 (182) | week | 912 |

approximates the original signal. The decomposition quality is influenced by the number of modal components. If too few modes are used, important signal information may be filtered out, reducing prediction accuracy. If too many modes are chosen, adjacent modes may have similar frequencies, causing redundancy or noise. Thus, selecting the appropriate number of modes is crucial.

The main difference between modes is their central frequency. Therefore, the number of modes can be determined by examining the distribution of these frequencies. Additionally, other VMD parameters are set as follows: the secondary penalty factor $\alpha$ is 2000, the noise tolerance $\tau$ is 0 to ensure decomposition fidelity, the central frequencies $\omega$ are randomly initialized using a uniform distribution, and the precision $\varepsilon$ is set to $1 \times 10^{-7}$. The center frequency values of each modal component of the CBFI at different K values are shown in Table 3. By a similar method, the modal component K values of the other three shipping indices, CCFI, SCFI and CDBI, can be obtained as 13, 9 and 10 respectively.

As shown in Table 3, when K of the CBFI is small (e.g., K=2), the frequency values of the modes are relatively large, and the distribution is wide, which may indicate that the modes contain more frequency components, lacking sufficient precision. As K increases, the frequency values of the modes gradually decrease, and the distribution becomes more refined, indicating that the modes begin to capture finer frequency components. When K is between 10 and 14, the frequency values of the modes stabilize, and the distribution becomes more detailed, suggesting that the optimal K value may lie within this range. Considering the balance between the number of modes and their refinement, this paper selects a K value of 10.

From Fig 4, we decompose the time series data of CBFI into 10 IMF components by using the VMD method. This decomposition method helps us gain a deeper understanding of the different frequency components in the data and their corresponding dynamic characteristics. Below is a detailed description of each IMF component: (1) imf1: The first IMF component typically contains the lowest frequency components, showing a slowly varying trend. This is the slowest changing part of the time series, possibly representing long-term trends. (2) imf2 to imf5: These IMF components contain medium-frequency components. As we move from imf2 to imf5, the frequency of fluctuations gradually increases, and the amplitude may first increase and then decrease. These components likely reflect medium-term fluctuations in the time series, such as certain periodic changes. (3) imf6 to imf9: These IMF components may contain higher frequency components, with smaller amplitudes and faster variations. They are likely to correspond to short-term fluctuations or high-frequency noise in the time series. (4) imf10: The tenth IMF component typically contains the highest frequency components, exhibiting characteristics of high-frequency noise or rapid changes. This is the fastest changing part of the time series, possibly corresponding to random noise or very short-period fluctuations.

**Table 3**. The center frequency values of each modal component of the CBFI under different K values.

| K | IMF1 | IMF2 | IMF3 | IMF4 | IMF5 | IMF6 | IMF7 | IMF8 | IMF9 | IMF10 | IMF11 | IMF12 | IMF13 |
|---|------|------|------|------|------|------|------|------|------|-------|-------|-------|-------|
| 2 | 95.6574 | — | — | — | — | — | — | — | — | — | — | — | — |
| 3 | 0.1667 | 0.3333 | — | — | — | — | — | — | — | — | — | — | — |
| 4 | 0.1250 | 0.2500 | 0.3750 | — | — | — | — | — | — | — | — | — | — |
| 5 | 0.1000 | 0.2000 | 0.3000 | 0.4000 | — | — | — | — | — | — | — | — | — |
| 6 | 0.0833 | 0.1667 | 0.2500 | 0.3333 | 0.4167 | — | — | — | — | — | — | — | — |
| 7 | 0.0714 | 0.1429 | 0.2143 | 0.2857 | 0.3571 | 0.4286 | — | — | — | — | — | — | — |
| 8 | 0.0625 | 0.125 | 0.1875 | 0.2500 | 0.3125 | 0.3750 | 0.4375 | — | — | — | — | — | — |
| 9 | 0.0556 | 0.1111 | 0.1667 | 0.2222 | 0.2778 | 0.3333 | 0.3889 | 0.4444 | — | — | — | — | — |
| 10 | 0.0500 | 0.1000 | 0.1500 | 0.2000 | 0.2500 | 0.3000 | 0.3500 | 0.4000 | 0.4500 | — | — | — | — |
| 11 | 0.0455 | 0.0909 | 0.1364 | 0.1818 | 0.2273 | 0.2727 | 0.3182 | 0.3636 | 0.4091 | 0.4545 | — | — | — |
| 12 | 0.0417 | 0.0833 | 0.1250 | 0.1667 | 0.2083 | 0.2500 | 0.2917 | 0.3333 | 0.375 | 0.4167 | 0.4583 | — | — |
| 13 | 0.0385 | 0.0769 | 0.1154 | 0.1538 | 0.1923 | 0.2308 | 0.2692 | 0.3077 | 0.3462 | 0.3846 | 0.4231 | 0.4615 | — |
| 14 | 0.0357 | 0.0714 | 0.1071 | 0.1429 | 0.1786 | 0.2143 | 0.2500 | 0.2857 | 0.3214 | 0.3571 | 0.3929 | 0.4286 | 0.4643 |

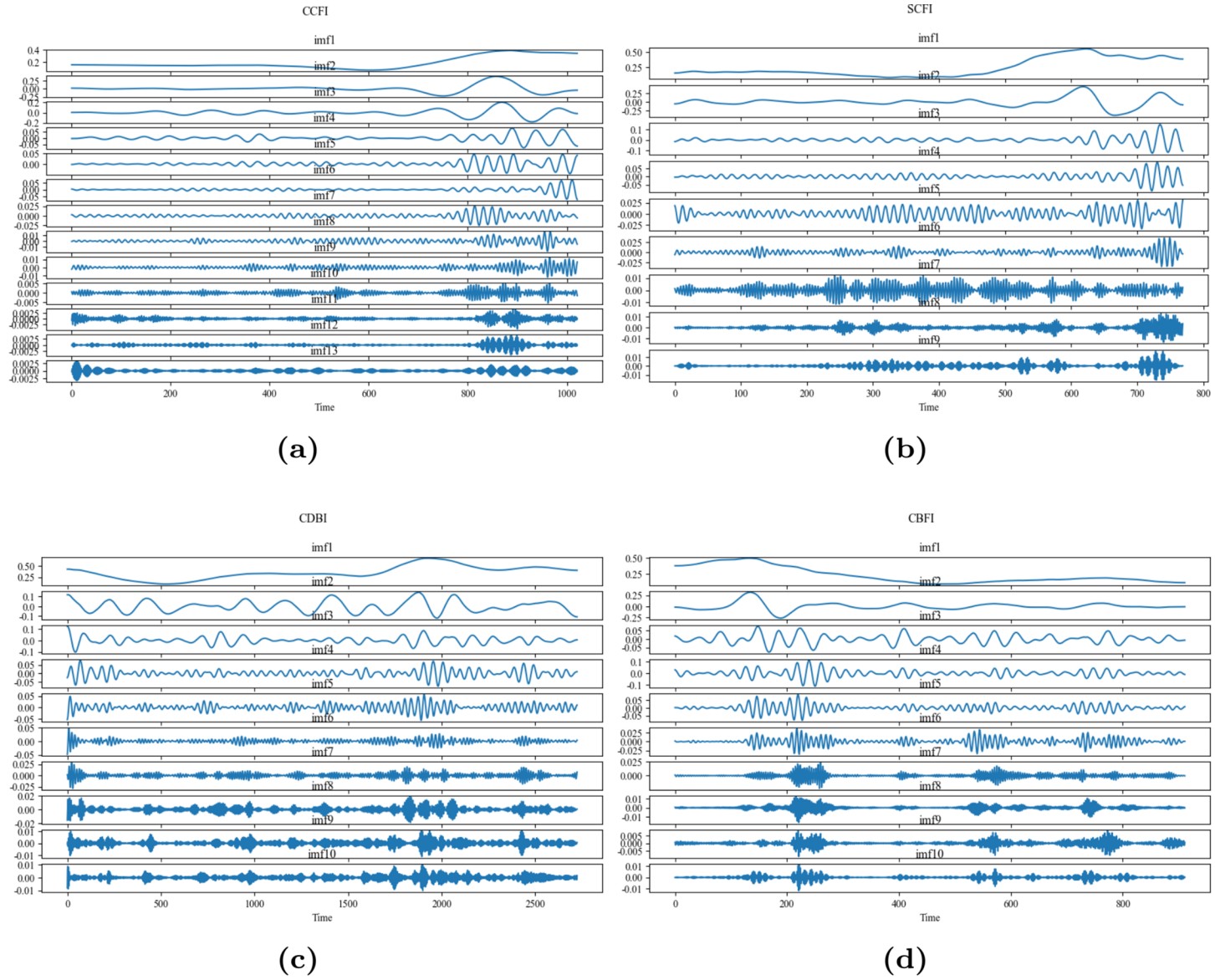

**Fig 4. VMD decomposition of China's shipping indices:** (**a**) VMD decomposition of CCFI. (**b**) VMD decomposition of SCFI. (**c**) VMD decomposition of CDBI. (**d**) VMD decomposition of CBFI.

After the VMD decomposition, the CBFI index is divided into 10 IMF components. We use the last 3 IMF components to reconstruct the signal and demonstrate the denoising effect. As shown in the Fig 5, the denoising effect of the variational mode decomposition indicates that, compared to the original signal, the reconstructed signal is smoother, with most of the noise effectively removed. The original and reconstructed signals remain consistent in their main trends.

To evaluate whether the denoising method successfully preserves the main features and trends of the signal while effectively removing noise, the Pearson correlation coefficient is used to measure the correlation between the original signal and the denoised signal. The correlation coefficient ranges from −1 to 1, with values closer to 1 indicating that the denoised signal is highly consistent with the original signal in terms of structure and trend. This suggests that the denoising method effectively removes noise while retaining important information, thus better reflecting the true pattern of the

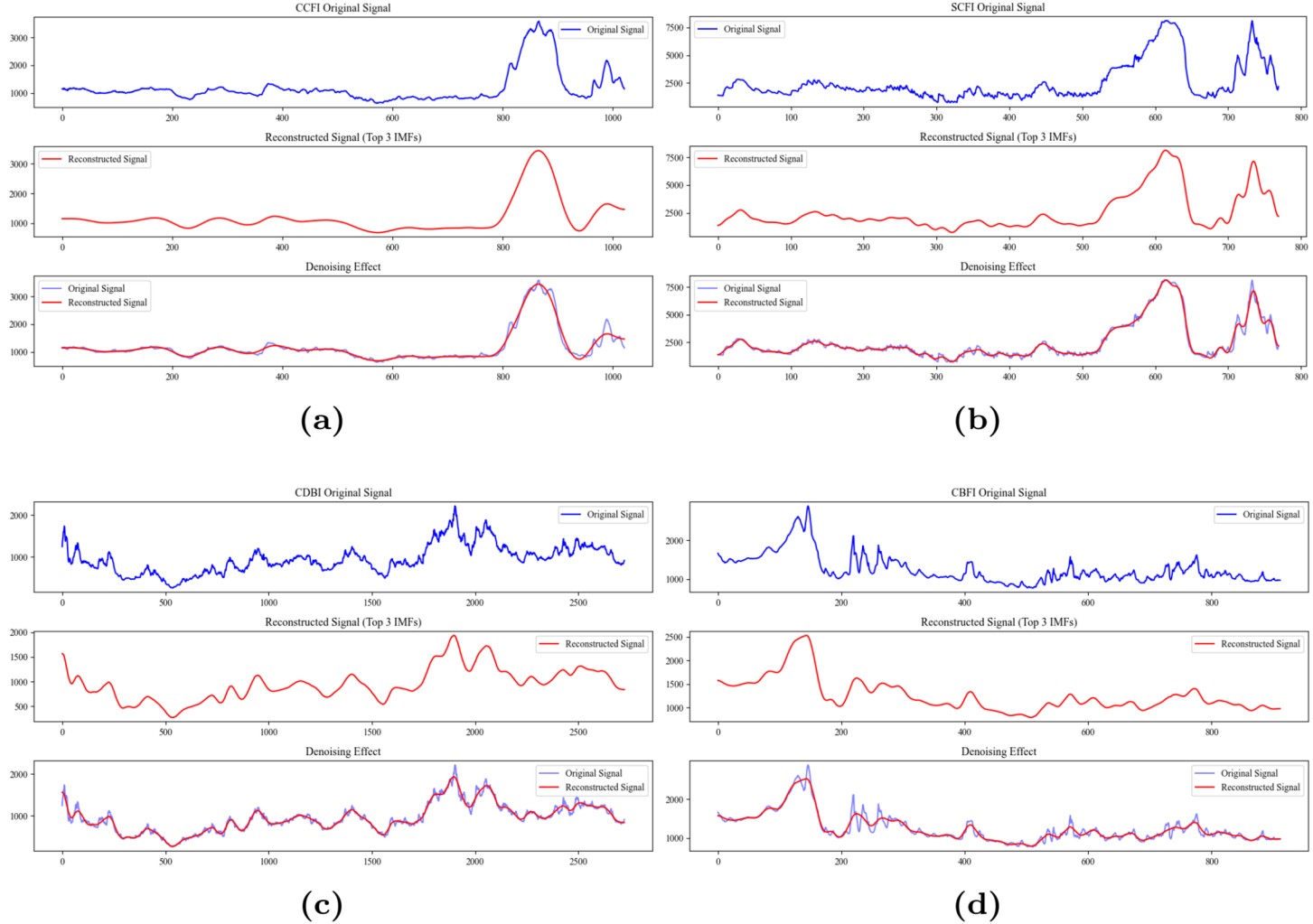

**Fig 5**. **The VMD decomposition recomposition of China's shipping indices:** (**a**) The VMD decomposition recomposition of CCFI. (**b**) The VMD decomposition recomposition of SCFI. (**c**) The VMD decomposition recomposition of CDBI. (**d**) The VMD decomposition recomposition of CBFI.

signal. Conversely, a correlation coefficient closer to −1 indicates significant differences between the signals, implying poor denoising performance. Compared to the MSE metric, the Pearson correlation coefficient more accurately reflects the impact of the denoising method on the signal's local features, as it takes into account the variations in different frequency components of the signal. The formula for calculating the Pearson correlation coefficient is as follows:

$$\rho(X, Y) = \frac{cov(X, Y)}{\sigma_X \sigma_Y}, \tag{14}$$

where $cov(X, Y)$ is the covariance between $X$ and $Y$. $\sigma_X$ and $\sigma_Y$ denote the standard deviations of $X$ and $Y$.

Based on the results in Table 4, it can be observed that both the EMD and VMD decomposition methods effectively remove noise from the signal. However, from the perspective of error metrics, the values of MSE and MAE for VMD decomposition is smaller than those of EMD decomposition, which implies that the VMD decomposition outperforms the EMD decomposition. In terms of the Pearson correlation coefficient $\rho$, the correlation coefficients of all shipping indices

**Table 4**. Comparison of the decomposition effects of VMD and EMD.

| Indices | VMD | | | EMD | | |
|---------|-----|-----|-----|-----|-----|-----|
| | MSE | MAE | $\rho$ | MSE | MAE | $\rho$ |
| CCFI | 4543.750 | 43.946 | 0.936 | 23804.320 | 285.690 | 0.602 |
| SCFI | 3743.990 | 136.390 | 0.969 | 13257.994 | 676.920 | 0.739 |
| CDBI | 3522.120 | 42.873 | 0.981 | 22210.210 | 114.520 | 0.898 |
| CBFI | 6371.011 | 53.388 | 0.952 | 8456.125 | 95.506 | 0.929 |

decomposed by VMD are above 0.95, while the correlation coefficients of all shipping indices decomposed by VMD are less than 0.95. The closer the value is to 1, the stronger the correlation, indicating that the VMD decomposition method performs better and better preserves the features of the original signal.

## Prediction results

**VMD-BiLSTM prediction.** The components obtained from the VMD decomposition are used as the input layer for the BiLSTM model. The model is configured with 2 hidden layers, each containing 50 neurons. The Adam optimizer is used, with the loss function set to Mean Squared Error (MSE). The number of epochs is set to 100, and the batch size is set to 32. 80% of the dataset is used for training, and 20% for testing. The prediction step length is set to 10, meaning that predictions are made for the 10 modal components obtained from the VMD decomposition, and the individual predictions are summed to obtain the final forecasted shipping index. To progressively capture trends and patterns in the data, and adapt to variations at different time points, thus enhancing the flexibility and accuracy of predictions, this study employs a step-wise prediction approach. Multi-step forecasting provides a more rigorous assessment of its capacity to capture long-term temporal dependencies and resist error propagation. By observing how errors scale with increasing prediction horizons, one can gain critical insights into the model's stability and robustness.

As shown in Table 5, under the same conditions, the prediction error of EMD-BiLSTM is greater than that of VMD-BiLSTM, and its coefficient of determination is lower. This is mainly because EMD is susceptible to noise interference during the decomposition process, and its recursive and locally adaptive characteristics may cause patterns in complex or nonlinear time series to become blurred, which affects the accuracy of the prediction. In contrast, VMD performs global optimization through variational principles, making the decomposition process more robust. It effectively reduces the impact of noise and improves the model's prediction accuracy, resulting in more accurate predictions and a higher coefficient of determination. Fig 6 shows that the 1-step prediction curve is closer to the real curve compared to the 3-step and 6-step prediction curves, and as indicated in Table 5, the errors of the 1-step prediction are smaller than those of the 3-step and 6-step predictions. This is because the 1-step prediction relies solely on the current known data, avoiding error accumulation based on previous predictions, which ensures a lower error. On the other hand, multi-step predictions rely

**Table 5**. The prediction results of China's shipping indices based on the EMD-BiLSTM and VMD-BiLSTM models.

| Models | Indices | RMSE | | | MAE | | | MPSE | | | $R^2$ | | |
|--------|---------|--------|--------|--------|--------|--------|--------|--------|--------|--------|--------|--------|--------|
| | | 1-step | 3-step | 6-step | 1-step | 3-step | 6-step | 1-step | 3-step | 6-step | 1-step | 3-step | 6-step |
| EMD-BiLSTM | CCFI | 153.132 | 147.452 | 164.715 | 122.324 | 118.573 | 145.712 | 10.758 | 12.378 | 13.621 | 0.948 | 0.943 | 0.940 |
| | SCFI | 532.124 | 743.723 | 993.013 | 387.321 | 542.631 | 743.214 | 12.782 | 16.352 | 22.821 | 0.949 | 0.889 | 0.812 |
| | CDBI | 14.376 | 15.243 | 18.834 | 9.513 | 11.268 | 16.137 | 0.735 | 0.956 | 1.685 | 0.983 | 0.978 | 0.963 |
| | CBFI | 54.354 | 19.786 | 27.441 | 10.327 | 16.234 | 20.933 | 0.946 | 1.964 | 1.995 | 0.975 | 0.968 | 0.954 |
| VMD-BiLSTM | CCFI | 137.435 | 139.317 | 148.750 | 101.427 | 106.850 | 120.680 | 8.572 | 8.650 | 9.261 | 0.977 | 0.977 | 0.974 |
| | SCFI | 516.654 | 733.475 | 981.005 | 366.060 | 517.740 | 725.660 | 10.952 | 14.021 | 19.686 | 0.951 | 0.900 | 0.821 |
| | CDBI | 9.802 | 12.238 | 16.729 | 7.215 | 9.186 | 13.025 | 0.627 | 0.802 | 1.149 | 0.996 | 0.993 | 0.987 |
| | CBFI | 10.067 | 16.701 | 23.305 | 8.274 | 13.064 | 18.339 | 0.726 | 1.131 | 1.614 | 0.995 | 0.986 | 0.974 |

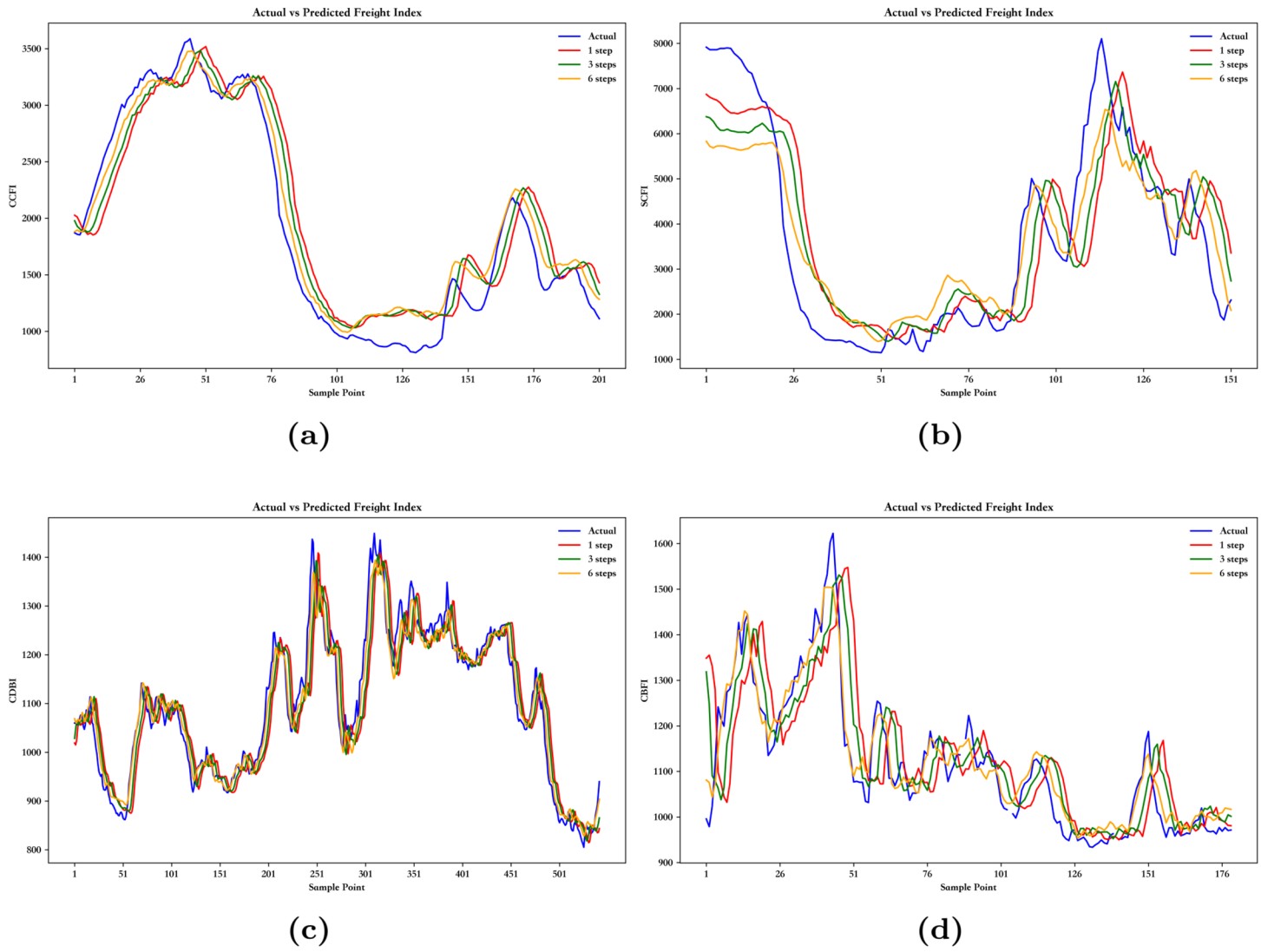

**Fig 6. The multi-step prediction curves of China's shipping indices based on VMD-BiLSTM model:** (**a**) The multi-step prediction curves of CCFI. (**b**) The multi-step prediction curves of SCFI. (**c**) The multi-step prediction curves of CDBI. (**d**) The multi-step prediction curves of CBFI.

on the results of previous steps, and errors propagate and accumulate with each step, leading to an increase in the final prediction error. Therefore, 1-step prediction is generally more effective and accurate.

**CPSO-BiLSTM model prediction.** The process of CPSO-BiLSTM for predicting shipping indices involves using the CPSO algorithm to optimize the hyperparameters of the BiLSTM model, thereby improving the model's training efficiency and prediction accuracy. Specifically, the CPSO algorithm optimizes the learning rate and other hyperparameters of the BiLSTM within the given parameter range (lb = [0.001, 10], ub = [0.01, 100]). The main parameters of CPSO are set as follows: cognitive factor $c_1 = 1.5$, social factor $c_2 = 1.7$, random factors $r_1 = 0.8$, $r_2 = 0.3$, and inertia weight $\omega = 0.5$, which control the particle's search behavior and enhance search efficiency. The BiLSTM model's parameters include: batch size $batch = 32$ and training epochs $epoch = 100$, where the number of epochs determines the number of training iterations,

and the batch size affects the number of samples used in each training step. By optimizing the BiLSTM's hyperparameters using the CPSO algorithm, the accuracy and training efficiency of the shipping index prediction can be significantly improved.

As shown in Table 6 and Fig 7, the prediction errors of PSO-BiLSTM is greater than that of CPSO-BiLSTM under the same setting, and its coefficient of determination is smaller than that of CPSO-BiLSTM, indicating that the prediction results of the CPSO-BiLSTM model are more accurate. This is mainly because the introduction of chaotic maps in CPSO enables dynamic adjustment of search parameters, which effectively addresses premature convergence and substantially improves global search performance. Furthermore, the one-step prediction error based on CPSO-BiLSTM is smaller than the errors of the three-step and six-step predictions because one-step prediction relies only on the currently known data, avoiding the impact of error accumulation in multi-step predictions, thus ensuring more accurate predictions. It is worth noting that the CPSO-BiLSTM exhibits higher errors for the CDBI index than PSO-BiLSTM, which can be attributed to CDBI's daily-frequency nature (Table 2) versus other weekly indices. The high-frequency characteristics amplify error propagation in CPSO-BiLSTM, while PSO-BiLSTM's simpler architecture inadvertently provides noise smoothing.

**VMD-CPSO-BiLSTM model prediction.** The process of using CPSO-BiLSTM to predict shipping indices after VMD is as follows: First, the first 80% of the original shipping index sequence is used as the training set, and the remaining 20% as the test set. Then, VMD is applied to decompose the training set data into several mode components. Next, the CPSO algorithm is used to optimize the hyperparameters of the BiLSTM model to improve prediction accuracy. The main parameters of CPSO are set as follows: cognitive factor $c_1 = 1.5$, social factor $c_2 = 1.7$, random factors $r_1 = 0.8$, $r_2 = 0.3$, and inertia weight $\omega = 0.5$. For the BiLSTM model, $unit = 50$, indicating that the number of units in the hidden layer is 50; $batch = 32$, meaning the batch size used in each training step is 32; $epoch = 100$, which determines the number of training epochs. Through this method, the original sequence is first decomposed using VMD to extract features, then the CPSO algorithm optimizes the BiLSTM model's hyperparameters, and finally, the shipping index is accurately predicted.

From Table 7 and Fig 8, it can be seen that the prediction performance of VMD-CPSO-BiLSTM is superior to that of VMD-PSO-BiLSTM. This is primarily because VMD-CPSO-BiLSTM combines the advantages of VMD, the CPSO algorithm, and BiLSTM networks. VMD effectively decomposes the time series signal into multiple frequency bands, allowing each component to better capture the local features and trends in the data. Meanwhile, CPSO, as an optimization algorithm, can more accurately adjust the parameters of the BiLSTM in a larger search space, overcoming the local optima problem of traditional particle swarm optimization, thus improving the model's prediction accuracy. Additionally, consistent with the previous conclusion on multi-step predictions, the multi-step prediction based on VMD-CPSO-BiLSTM still shows that 1-step prediction outperforms the 3-step and 6-step predictions.

**Prediction result analysis.** To verify the advantages of the BiLSTM model, this subsection compares the prediction errors and determination coefficients of the CNN, LSTM and BiLSTM models. Additionally, to assess the ability of CPSO

**Table 6. The prediction results based on the PSO-BiLSTM and CPSO-BiLSTM models.**

| Models | Indices | RMSE | | | MAE | | | MPSE | | | $R^2$ | | |
|---|---|---|---|---|---|---|---|---|---|---|---|---|---|
| | | 1-step | 3-step | 6-step | 1-step | 3-step | 6-step | 1-step | 3-step | 6-step | 1-step | 3-step | 6-step |
| PSO-BiLSTM | CCFI | 183.561 | 206.647 | 257.237 | 153.52 | 187.527 | 231.583 | 12.318 | 11.039 | 13.696 | 0.959 | 0.949 | 0.922 |
| | SCFI | 623.226 | 651.707 | 873.472 | 419.949 | 432.567 | 617.044 | 11.318 | 13.232 | 15.063 | 0.928 | 0.921 | 0.858 |
| | CDBI | 12.648 | 14.845 | 17.616 | 9.211 | 10.237 | 14.145 | 0.812 | 0.947 | 1.307 | 0.993 | 0.972 | 0.956 |
| | CBFI | 21.112 | 23.325 | 31.162 | 17.412 | 18.931 | 24.995 | 1.549 | 1.674 | 2.197 | 0.979 | 0.975 | 0.954 |
| CPSO-BiLSTM | CCFI | 166.191 | 192.370 | 245.630 | 123.981 | 176.234 | 220.456 | 10.228 | 11.345 | 13.023 | 0.961 | 0.958 | 0.932 |
| | SCFI | 439.290 | 521.690 | 781.081 | 309.382 | 329.882 | 546.245 | 9.236 | 7.273 | 13.271 | 0.929 | 0.909 | 0.886 |
| | CDBI | 10.298 | 15.023 | 22.034 | 8.221 | 12.045 | 18.067 | 0.776 | 1.234 | 1.856 | 0.993 | 0.964 | 0.952 |
| | CBFI | 9.901 | 14.892 | 19.414 | 7.421 | 11.210 | 15.340 | 0.406 | 0.979 | 1.361 | 0.985 | 0.971 | 0.950 |

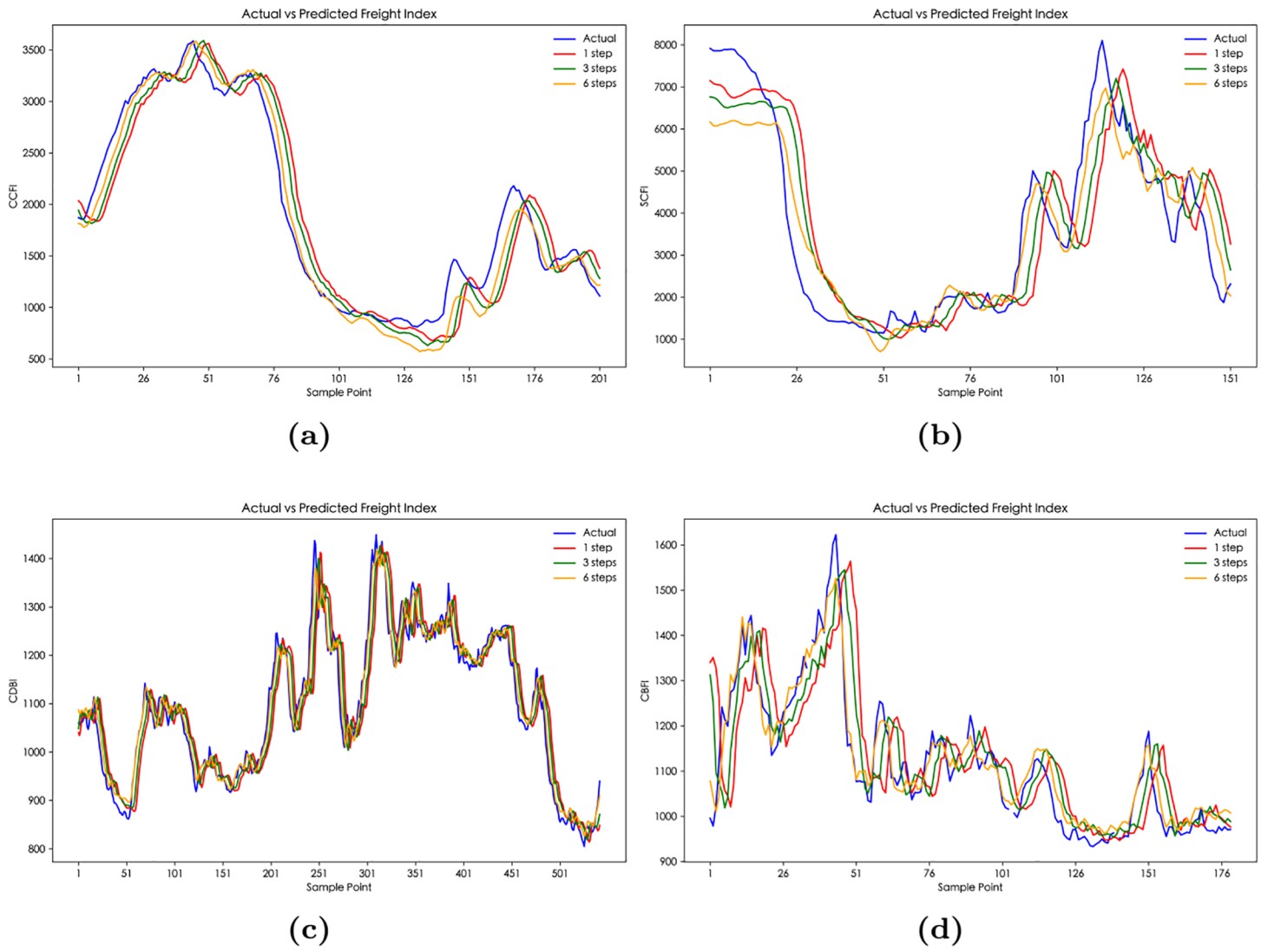

**Fig 7**. **The multi-step prediction curves of China's shipping indices based on CPSO-BiLSTM model:** (**a**) The multi-step prediction curves of CCFI. (**b**) The multi-step prediction curves of SCFI. (**c**) The multi-step prediction curves of CDBI. (**d**) The multi-step prediction curves of CBFI.

**Table 7**. **The prediction results of China's shipping indices based on the VMD-PSO-BiLSTM and VMD-CPSO-BiLSTM models.**

| Models | Indices | RMSE | | | MAE | | | MPSE | | | $R^2$ | | |
|---|---|---|---|---|---|---|---|---|---|---|---|---|---|
| | | 1-step | 3-step | 6-step | 1-step | 3-step | 6-step | 1-step | 3-step | 6-step | 1-step | 3-step | 6-step |
| VMD-PSO-BiLSTM | CCFI | 80.283 | 103.446 | 156.16 | 62.201 | 74.056 | 118.812 | 4.201 | 5.797 | 8.916 | 0.990 | 0.987 | 0.971 |
| | SCFI | 265.394 | 415.566 | 654.283 | 171.372 | 267.294 | 462.096 | 4.289 | 5.897 | 11.358 | 0.989 | 0.968 | 0.92 |
| | CDBI | 4.093 | 6.301 | 8.583 | 2.387 | 3.313 | 6.712 | 0.199 | 0.301 | 0.612 | 0.998 | 0.994 | 0.990 |
| | CBFI | 5.242 | 15.110 | 21.057 | 4.0701 | 11.458 | 16.580 | 0.358 | 0.984 | 1.433 | 0.998 | 0.989 | 0.979 |
| VMD-CPSO-BiLSTM | CCFI | 67.096 | 97.159 | 140.92 | 45.365 | 68.921 | 105.129 | 3.77 | 5.496 | 7.968 | 0.994 | 0.989 | 0.976 |
| | SCFI | 193.523 | 346.547 | 595.754 | 135.988 | 257.321 | 466.889 | 3.476 | 6.928 | 13.063 | 0.993 | 0.978 | 0.934 |
| | CDBI | 2.546 | 4.078 | 7.821 | 1.957 | 3.086 | 6.056 | 0.179 | 0.284 | 0.555 | 0.999 | 0.999 | 0.997 |
| | CBFI | 4.949 | 16.206 | 20.878 | 3.703 | 12.065 | 16.049 | 0.326 | 1.039 | 1.377 | 0.998 | 0.988 | 0.979 |

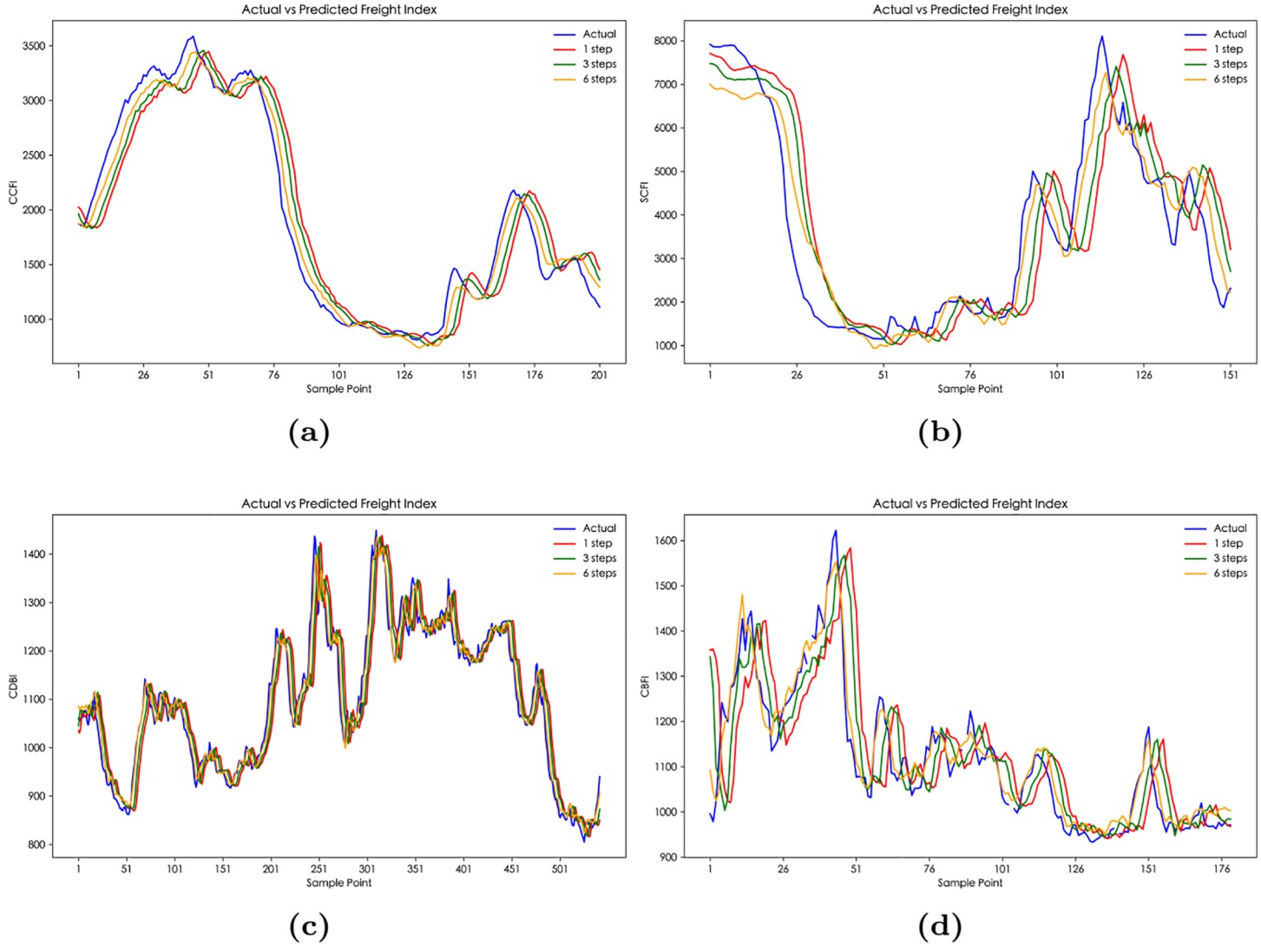

**Fig 8. The multi-step prediction curves of China's shipping indices based on VMD-CPSO-BiLSTM model:** (**a**) The multi-step prediction curves of CCFI. (**b**) The multi-step prediction curves of SCFI. (**c**) The multi-step prediction curves of CDBI. (**d**) The multi-step prediction curves of CBFI.

to optimize the hyperparameters of the models, the paper compares models optimized by PSO (PSO-BiLSTM) and models optimized by CPSO (CPSO-BiLSTM). To validate the effectiveness of the VMD decomposition method in shipping index prediction, this paper also decomposes the original sequence using the VMD method and constructs six combination models for comparison (VMD-BiLSTM, VMD-PSO-BiLSTM, VMD-CPSO-BiLSTM). Through these eight models, the goal is to demonstrate the advantages of the VMD-CPSO-BiLSTM model proposed in this paper.

The following conclusions can be drawn from Figs 9–11 and Table 8: (1) Through the comparison of single models, it is evident that the BiLSTM model has the best prediction performance, while the CNN performs the worst. This is mainly because BiLSTM, with its bidirectional structure, considers both past and future information, while CNN relies on local feature extraction, making it difficult to capture long-term dependencies.

(2) In terms of parameter optimization performance, CPSO is superior to PSO. The model optimized by CPSO achieves better prediction accuracy than the one optimized by PSO. That is, the prediction error of CPSO-BiLSTM is

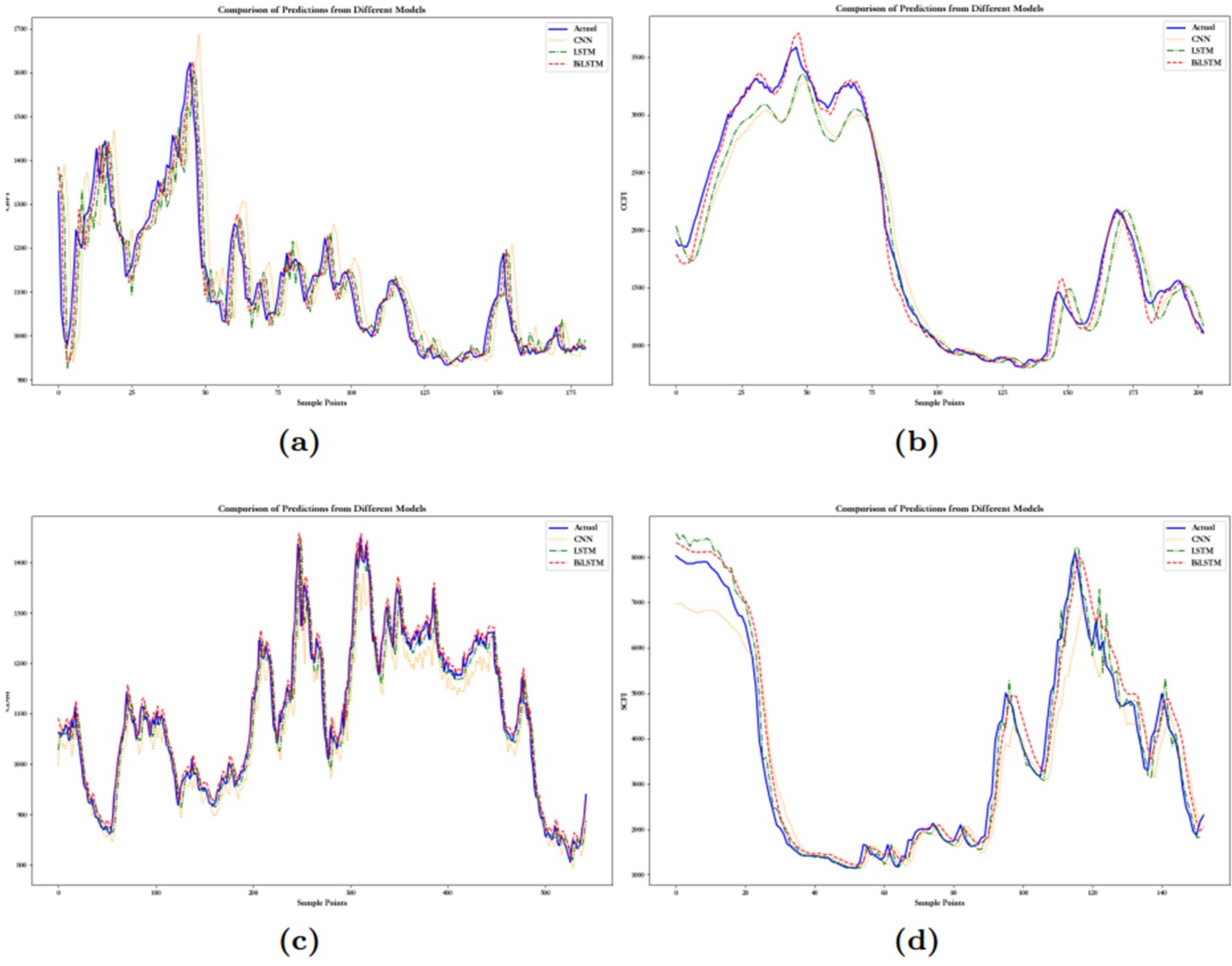

**Fig 9**. **The prediction curves of China's shipping indices based on several single models:** (**a**) The prediction curves of CBFI. (**b**) The prediction curves of CCFI. (**c**) The prediction curves of CDBI. (**d**) The prediction curves of SCFI.

smaller than that of PSO-BiLSTM, and the correlation coefficient is greater than that of PSO-BiLSTM. As CPSO combines a cooperative evolution strategy, allowing for more efficient exploration of the solution space and avoiding local optima.

(3) From the perspective of modal decomposition performance, the model after VMD decomposition outperforms the one without decomposition. The prediction accuracy of VMD-BiLSTM is better than that of BiLSTM, VMD-PSO-BiLSTM is better than PSO-BiLSTM, and VMD-CPSO-BiLSTM is better than CPSO-BiLSTM. Across the four indices, the RMSEs improvement of CPSO-BiLSTM over PSO-BiLSTM (9.46%, 29.53%, 18.60%, and 5.74%) is significantly amplified after introducing VMD, with the enhancement increasing to 16.43%, 39.19%, 37.80%, and 24.67%, respectively. Mainly because VMD can more precisely extract the intrinsic modes of the signal, reducing noise interference and improving pre-diction accuracy. In additional, the CEEMD-PSO-BiLSTM model demonstrates better prediction performance than PSO-BiLSTM model, though it underperforms compared to the VMD-PSO-BiLSTM model, not to mention being significantly

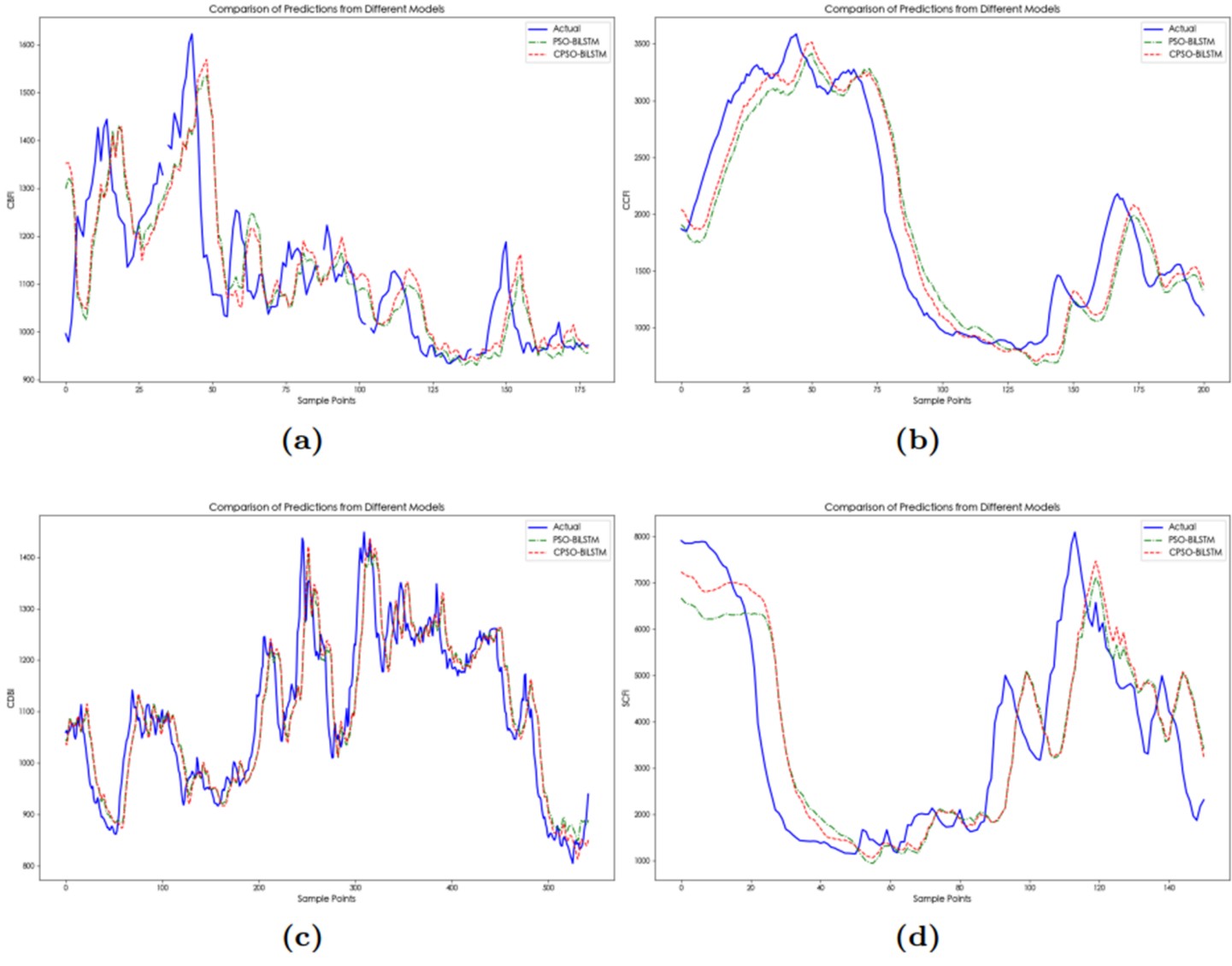

**Fig 10**. **The prediction curves of China's shipping indices based on several optimized models:** (**a**) The prediction curves of CBFI. (**b**) The prediction curves of CCFI. (**c**) The prediction curves of CDBI. (**d**) The prediction curves of SCFI.

outperformed by the VMD-CPSO-BiLSTM model. Therefore, Among all individual and combined models, the VMD-CPSO-BiLSTM combination demonstrates the strongest prediction performance, with the three prediction errors minimized and the coefficient of determination maximized.

## Conclusion and discussion

This paper proposes an innovative combined model, VMD-CPSO-BiLSTM, for predicting shipping indices, aiming to improve the prediction accuracy of indices in China's shipping market and provide more reliable support for shipping decision-making. Although [30] used the CEEMD-PSO-BiLSTM model to predict the shipping index. However, CEEMD may suffer from mode mixing, leading to feature loss or redundancy, while PSO is prone to being trapped in local minima, reducing optimization efficiency. In contrast, VMD avoids mode mixing through precise frequency-domain decomposition,

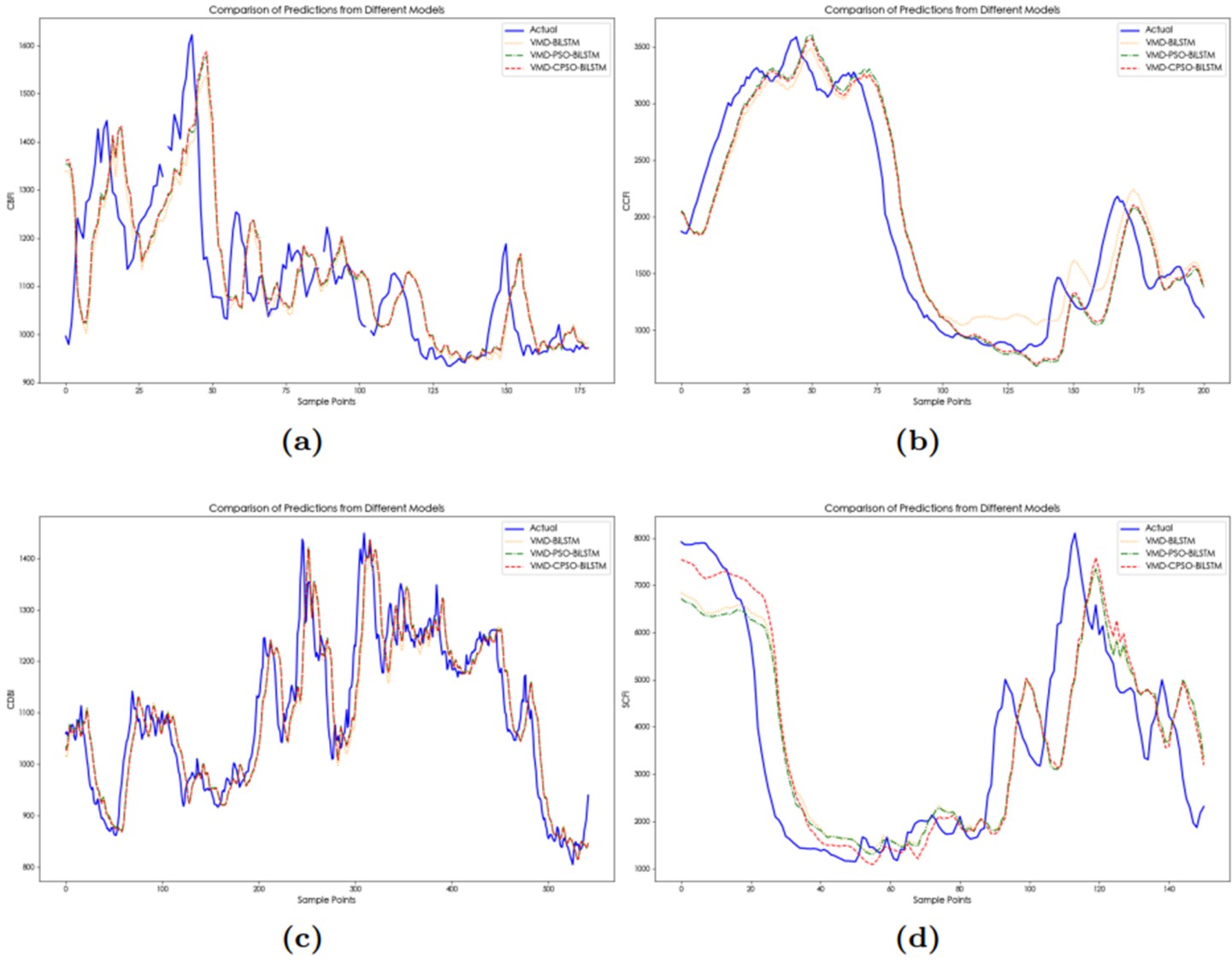

**Fig 11**. **The prediction curves of China's shipping indices based on several decomposed models:** (**a**) The prediction curves of CBFI. (**b**) The prediction curves of CCFI. (**c**) The prediction curves of CDBI. (**d**) The prediction curves of SCFI.

enabling clearer extraction of signal features and improving prediction accuracy. CPSO introduces a chaotic search mechanism, enhancing global search capability, avoiding local minima, and improving optimization efficiency. Therefore, the BiLSTM model optimized by VMD and CPSO can more effectively handle the nonlinearity and temporal dependencies of the shipping index, while improving prediction accuracy and ensuring training speed. Compared to the CEEMD-PSO model, it offers significant advantages and provides more reliable decision support for predicting the shipping index.

1. Interdisciplinary integration and method evolution.

In the field of shipping index prediction, the VMD-CPSO-BiLSTM model represents a new methodological breakthrough. By integrating VMD, CPSO and BiLSTM networks, the model achieves an organic combination of data decomposition, intelligent optimization, and deep learning. This provides a novel technical approach to handling complex,

**Table 8**. Comparison of prediction results of different models.

| Models | Indices | CCFI | | | SCFI | | |
|---|---|---|---|---|---|---|---|
| | | RMSE | MAE | $R^2$ | RMSE | MAE | $R^2$ |
| Single models | CNN | 224.252 | 175.88 | 0.941 | 330.5928 | 292.519 | 0.902 |
| | LSTM | 216.429 | 158.895 | 0.943 | 558.808 | 365.393 | 0.939 |
| | BiLSTM | 190.268 | 148.928 | 0.949 | 491.934 | 365.802 | 0.953 |
| Optimized | PSO-BiLSTM | 183.563 | 133.522 | 0.959 | 623.226 | 419.941 | 0.928 |
| | CPSO-BiLSTM | 166.191 | 123.980 | 0.961 | 439.298 | 309.38 | 0.929 |
| Decomposed | VMD-BiLSTM | 137.431 | 101.423 | 0.977 | 516.654 | 366.061 | 0.950 |
| | VMD-PSO-BiLSTM | 80.283 | 62.201 | 0.991 | 318.238 | 171.37 | 0.989 |
| | VMD-CPSO-BiLSTM | 67.096 | 45.365 | 0.994 | 193.523 | 135.980 | 0.993 |
| | CEEMD-PSO-BiLSTM | 106.342 | 94.753 | 0.985 | 383.706 | 202.643 | 0.945 |
| Models | Indices | CDBI | | | CBFI | | |
| | | RMSE | MAE | $R^2$ | RMSE | MAE | $R^2$ |
| Single models | CNN | 32.112 | 24.351 | 0.940 | 76.012 | 51.281 | 0.752 |
| | LSTM | 23.912 | 20.334 | 0.958 | 44.549 | 33.0498 | 0.898 |
| | BiLSTM | 19.853 | 14.629 | 0.982 | 43.424 | 29.628 | 0.917 |
| Optimized | PSO-BiLSTM | 12.641 | 9.210 | 0.992 | 21.113 | 17.411 | 0.979 |
| | CPSO-BiLSTM | 10.290 | 8.221 | 0.993 | 19.901 | 7.421 | 0.985 |
| Decomposed | VMD-BiLSTM | 9.801 | 7.215 | 0.995 | 10.061 | 8.273 | 0.995 |
| | VMD-PSO-BiLSTM | 4.093 | 2.387 | 0.998 | 5.242 | 4.070 | 0.997 |
| | VMD-CPSOBiLSTM | 2.546 | 1.957 | 0.999 | 3.949 | 3.703 | 0.998 |
| | CEEMD-PSO-BiLSTM | 6.217 | 4.506 | 0.994 | 8.435 | 5.963 | 0.989 |

nonlinear, and non-stationary data. This interdisciplinary model innovation not only enhances prediction accuracy but also offers valuable insights and references for time series prediction problems in other fields.

2. Accurate prediction and decision support.

As the global shipping industry faces increasingly complex market changes and economic conditions, precise shipping index predictions are of paramount importance for industry decision-making and policy formulation. The successful application of the VMD-CPSO-BiLSTM model provides shipping companies, investors, and policymakers with more scientifically grounded market trend forecasts, thereby optimizing shipping resource allocation, improving operational efficiency, and reducing market risks. In practical applications, this model serves as a data-driven decision support tool for the shipping industry, helping it navigate and respond to complex market challenges.

3. Forward-looking exploration of future research and technological evolution.

Integration of Deep Learning and Multi-Scale Data Processing In the future, as shipping market data continues to grow and technology advances, the VMD-CPSO-BiLSTM model will undergo further expansion and optimization on multiple levels. First, future research could explore how to integrate more advanced deep learning algorithms, such as Generative Adversarial Networks (GANs) and Reinforcement Learning (RL), to further enhance prediction accuracy. Additionally, the fusion of multi-source data, such as weather and port information, can provide more dimensional support for predicting shipping indices, improving the model's precision and stability in practical applications. Ultimately, the construction of intelligent shipping systems based on artificial intelligence and big data will be a key research direction, enabling the intelligent and automated development of the shipping industry.

## Suggestion

This paper predicts four important shipping indices in China (CCFI, SCFI, CDBI, and CBFI) using the VMD-CPSO-BiLSTM model, aiming to provide data support and decision-making basis for the formulation and development of policies in the shipping industry. CCFI, SCFI, CDBI and CBFI are important indicators for measuring the status of China's shipping

market and price fluctuations. By analyzing the trend of these indices, this paper proposes the following three policy recommendations to help the government and shipping companies respond to market changes and promote the sustainable development of the shipping industry.

1. Enhance shipping market monitoring and policy early warning mechanism.

Based on the predictions of the four shipping indices using the VMD-CPSO-BiLSTM model, the government should establish and improve a shipping market monitoring and early warning mechanism. By tracking the trends of CCFI, SCFI, CDBI, and CBFI in real-time, the government can identify market risks in advance and issue policy warnings promptly, providing guidance for shipping companies to help them manage risks and prepare for changes, thus reducing the negative impact of market fluctuations on the shipping industry.

2. Optimize shipping resource allocation and infrastructure development.

According to the model's predictions, the government should make accurate forecasts about changes in shipping market demand and allocate resources accordingly. For example, based on the predictions of SCFI and CCFI, the government should plan port facilities, channel construction, and container transportation networks in advance to meet the potential increase in shipping demand. By optimizing resource allocation, shipping efficiency can be improved, transportation costs reduced, and China's shipping industry's international competitiveness enhanced.

3. Promote green shipping and global cooperation.

As environmental protection requirements continue to strengthen, green shipping has become a key trend in the industry's development. The government can analyze the shipping index predictions in combination with global market changes to promote the formulation of green shipping policies and encourage shipping companies to invest in low-carbon and environmentally friendly technologies. Additionally, by utilizing changes in shipping indices, the government can enhance cooperation with international shipping markets, especially in dry bulk and container transport sectors, to promote the exchange and sharing of international shipping resources, thereby boosting China's shipping industry's global influence.

With the above policy recommendations, the government and shipping companies can better respond to market fluctuations and risks, optimize resource allocation, enhance international competitiveness, and promote the sustainable development of the shipping industry.

## Supporting information

**S1 Data.** https://pan.baidu.com/s/15QEj7tRDk5vlLM1SzPUNnA?pwd=mtks.
(XLSX)

**S1 Code.** https://pan.baidu.com/s/1ZlpBrF2K_cMXWohW43g43A?pwd=k8xd.
(ZIP)

## Acknowledgments

We thank the editor, an associate editor and there reviewers for their most helpful comments.

## Author contributions

**Investigation:** Guangnian XIAO.

**Methodology:** Yuye ZOU.

**Software:** Yingyu Liu.

**Writing – review & editing:** Yuye ZOU.

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
