## [Decision Letter · Decision Letter 0]

16 Jul 2025

PONE-D-25-29709Forecasting China's shipping indices based on modal decomposition and optimized deep learning integrated modelPLOS ONE

Dear Dr. ZOU,

Thank you for submitting your manuscript to PLOS ONE. After careful consideration, we feel that it has merit but does not fully meet PLOS ONE’s publication criteria as it currently stands. Therefore, we invite you to submit a revised version of the manuscript that addresses the points raised during the review process.

**Please address all the comments in the revised manuscript and a proof reading is highly recommended for the resubmission.**

We look forward to receiving your revised manuscript.

Kind regards,

Qichun Zhang, PhD

Academic Editor

PLOS ONE

Journal Requirements:

“National Natural Science Foundation of China (12101393�”

4. Please note that your Data Availability Statement is currently missing the repository name and/or the DOI/accession number of each dataset OR a direct link to access each database. If your manuscript is accepted for publication, you will be asked to provide these details on a very short timeline. We therefore suggest that you provide this information now, though we will not hold up the peer review process if you are unable.

5. Please ensure that you refer to Figure 1 and 2 in your text as, if accepted, production will need this reference to link the reader to the figure.

Additional Editor Comments:

A number of the comments have been received for this submission. More details are needed to address all these comments. Thus a major revision is essential for this paper.

Reviewers' comments:

Reviewer's Responses to Questions

**Comments to the Author**

1. Is the manuscript technically sound, and do the data support the conclusions?

Reviewer #1: Yes

Reviewer #2: Yes

2. Has the statistical analysis been performed appropriately and rigorously?

Reviewer #1: Yes

Reviewer #2: Yes

3. Have the authors made all data underlying the findings in their manuscript fully available?

Reviewer #1: Yes

Reviewer #2: Yes

4. Is the manuscript presented in an intelligible fashion and written in standard English?

Reviewer #1: Yes

Reviewer #2: Yes

5. Review Comments to the Author

Reviewer #1: Below are my suggestions for improvement.

About Manuscript Structure

1. Inclusion of a Dedicated Introduction Section: The manuscript currently moves directly from the Abstract to the Materials and Methods. I strongly recommend adding a distinct "Introduction" section.

2. Logical Ordering of Methodological Concepts: Your workflow is presented as VMD -> CPSO -> BiLSTM, but the introduction of these concepts in the introductory parts of the paper does not follow this logical order, with VMD being mentioned last. To improve clarity, I suggest reorganizing the introduction of these methods to mirror the order in which they are actually applied in your research. This will allow readers to follow your methodological approach more intuitively.

3. Placement of "Related Work" Comparison: You provide a comparison with a similarly structured model, CEEMD-PSO-BiLSTM, in the Conclusion and Discussion section. I would advise moving this discussion to the new "Introduction" section, or potentially creating a "Related Work" section. Placing this comparison earlier in the manuscript will help to highlight the novelty and contribution of your proposed model right from the start.

About Presentation

1. Standardize Abbreviations: On first use, define all abbreviations using the format "Full Name (Abbreviation)". Use only the abbreviation thereafter. A thorough check of the entire paper is required.

2. Illustrate the VMD Process: Add a flowchart or a formal algorithm to clearly illustrate its procedural steps.

3. Define Equation Notations: Immediately following every equation, you must explicitly define all mathematical notations used. Apply this consistently across the entire paper.

4. Unify Index Order: The presentation order of your four indices is inconsistent. In lines 304-314, you discuss CDBI before SCFI, which contradicts the CCFI, SCFI, CDBI, CBFI order used elsewhere. Please correct this for consistency.

5. Clarify Figure and Table Elements: Before analyzing any data in figures or tables, define all non-standard terms, abbreviations, and units within the caption or main text (e.g., explain ADF and the meaning of the p-value before discussing Table 1).

6. Unify Data Formatting: Ensure consistent data formatting within and across tables. For example, the decimal places for corresponding items in Table 5 and Table 6 should be uniform.

7. Correct Cross-References: Check the entire manuscript to ensure all your cross-references are in the right format.

About Experiments and Results Analysis

1. Inconsistent Units in Table 2: The unit for CDBI in Table 2 differs from other indices. Justify this discrepancy.

2. Justify Multi-Step Prediction: The comparison between 1-step and multi-step predictions currently rests only on accuracy. What is the motivation for including multi-step analysis? If it offers other advantages (e.g., efficiency, generality), analyze them. Otherwise, its purpose is unclear.

3. Explain Anomalous Results in Table 6: In your multi-step predictions (Table 6), the CPSO-BiLSTM model yields higher errors for the CDBI index than the standard PSO-BiLSTM. Explain this result.

4. Redundant Comparison and Potential Error: You have already established that CPSO-BiLSTM outperforms PSO-BiLSTM. Therefore, the comparison between VMD-CPSO-BiLSTM and VMD-PSO-BiLSTM seems redundant. What is the scientific justification for this experiment? Furthermore, lines 513-514 claim VMD-PSO-BiLSTM is superior to VMD-CPSO-BiLSTM, which directly contradicts your other findings. Is this a clerical error? Clarify the logic and correct any mistakes.

5. Add Essential Comparative Experiment: Your paper discusses the CEEMD-PSO-BiLSTM model. A direct experimental comparison with your proposed model is essential to validate your claims of superiority. This experiment should be conducted and included.

Reviewer #2: The abstract clearly presents a hybrid forecasting model, VMD-CPSO-BiLSTM, designed specifically for predicting China's shipping indices.

1. What is the difference between CPSO and PSO?

2. Need to add Justification for CPSO over PSO.

3. How long VMD, CPSO, BiLSTM takes (Time)

6. PLOS authors have the option to publish the peer review history of their article (what does this mean?). If published, this will include your full peer review and any attached files.

Reviewer #1: No

Reviewer #2: No

---

## [Author Response · Author response to Decision Letter 1]

26 Aug 2025

The authors have responded to the comments from the editor and reviewers, which has been uploaded as an attachment.

---

## [Decision Letter · Decision Letter 1]

2 Nov 2025

Forecasting China's shipping indices based on modal decomposition and optimized deep learning integrated model

PONE-D-25-29709R1

Dear Dr. ZOU,

We’re pleased to inform you that your manuscript has been judged scientifically suitable for publication and will be formally accepted for publication once it meets all outstanding technical requirements.

Kind regards,

Qichun Zhang, PhD

Academic Editor

PLOS ONE

Reviewers' comments:

Reviewer's Responses to Questions

**Comments to the Author**

1. If the authors have adequately addressed your comments raised in a previous round of review and you feel that this manuscript is now acceptable for publication, you may indicate that here to bypass the “Comments to the Author” section, enter your conflict of interest statement in the “Confidential to Editor” section, and submit your "Accept" recommendation.

Reviewer #2: All comments have been addressed

2. Is the manuscript technically sound, and do the data support the conclusions?

Reviewer #2: Yes

3. Has the statistical analysis been performed appropriately and rigorously?

Reviewer #2: Yes

4. Have the authors made all data underlying the findings in their manuscript fully available?

Reviewer #2: (No Response)

5. Is the manuscript presented in an intelligible fashion and written in standard English?

Reviewer #2: Yes

6. Review Comments to the Author

Reviewer #2: Thanks for the authors for addressed all comments and rewrite a good paper with good idea.

I recommended to accept this work.

7. PLOS authors have the option to publish the peer review history of their article (what does this mean?). If published, this will include your full peer review and any attached files.

Reviewer #2: No

---

## [Editor Report · Acceptance letter]

PONE-D-25-29709R1

PLOS ONE

Dear Dr. ZOU,

I'm pleased to inform you that your manuscript has been deemed suitable for publication in PLOS ONE. Congratulations! Your manuscript is now being handed over to our production team.

Kind regards,

on behalf of

Prof. Qichun Zhang

Academic Editor

PLOS ONE